# Preprocessing approaches in machine learning-based groundwater potential mapping: an application to the Koulikoro and Bamako regions, Mali

Víctor Gómez-Escalonilla [1*], Pedro Martínez-Santos[1], Miguel Martín-Loeches[2]

[1]UNESCO/UNITWIN Chair Appropriate Technologies for Human Development. Department of Geodynamic, Stratigraphy and Paleontology, Faculty of Geology, Complutense University of Madrid, C/José Antonio Novais 12, 28040 Madrid, Spain. [2]Department of Geology, Geography and Environment. Geology UD, University of Alcalá, Alcalá de Henares, Madrid, Spain.

*Correspondence to*: V. Gómez-Escalonilla (vigome01@ucm.es)

**Abstract.**

Groundwater is crucial for domestic supplies in the Sahel, where the strategic importance of aquifers will increase in the coming years due to climate change. Groundwater potential mapping is a valuable tool to underpin water management in the region, and hence, to improve drinking water access. This paper presents a machine learning method to map groundwater potential. This is illustrated through its application in two administrative regions of Mali. A set of explanatory variables for the presence of groundwater is developed first. Scaling methods (standardization, normalization, maximum absolute value and

min-max scaling) are used to avoid the pitfalls associated with reclassification. Noisy, collinear and counterproductive variables are identified and excluded from the input dataset. Twenty machine learning classifiers are then trained and tested on a large borehole database (n=3,345) in order to find meaningful correlations between the presence or absence of groundwater and the explanatory variables. Maximum absolute value and standardization proved the most efficient scaling techniques, while tree-based algorithms (accuracy >0.85) consistently outperformed other classifiers. The borehole flow rate

data was then used to calibrate the results beyond standard machine learning metrics, thus adding robustness to the predictions. The southern part of the study area presents the better groundwater prospect, this being consistent with the geological and climatic setting. Outcomes lead to three major conclusions: (1) picking the best performers out of a large number of machine learning classifiers is recommended as a good methodological practice; (2) standard machine learning metrics should be complemented with additional hydrogeological indicators whenever possible; and (3) variable scaling contributes to minimize

expert bias.

**Keywords:** Groundwater potentiality, GIS, random forest, decision trees, supervised classification

## 1 Introduction

Today, 2.5 billion people across the world depend on groundwater for domestic supply (Grönwall and Danert, 2020). Groundwater is crucial in arid regions, such as the Sahel, where aquifers provide a permanent source of good quality water during the months of the year in which rainfall and surface water are absent (Llamas and Martínez-Santos, 2005; Díaz-Alcaide et al., 2017). In a context of climate change, with rainfall expected to decrease and droughts likely to become more intense (Arneth et al., 2019), groundwater resources will be increasingly relied upon. This could well be the case in rural Mali, where access to drinking water and sanitation is a concern. In 2017, only 68% of the rural population had "at least basic" drinking water access, while 24% relied on unimproved water sources like unprotected springs and wells (UNICEF/WHO, 2019).

Groundwater potential mapping (GPM) is recognized as a valuable tool to underpin the planning and exploration of groundwater resources (Elbeih, 2015). There is no universal consensus as to what groundwater potential means. Thus, GPM may consist of developing spatial estimates of groundwater storage in a given region, measuring the probability of finding groundwater, or predicting where the highest borehole yields may occur (Díaz-Alcaide and Martínez-Santos, 2019). Recent years have witnessed a growing interest in groundwater potential studies across Africa, largely as a result of the need to achieve Sustainable Development Goal #6. The majority of these studies rely on a combination of remote sensing, geographic information systems and geophysics (Delgado 2018, Adeyeye et al., 2019, Magaia et al 2018, Mpofu et al 2020, Owolabi et al 2020, Saadi et al 2021, Al-Djazouli et al. 2021), while others directly interpret the information from borehole databases (Díaz-Alcaide et al 2017).

There are two main approaches to GPM, namely, expert-based decision systems and machine learning methods. Expert-based techniques have been used for a long time (DEP, 1993), and include multi-influence factor approaches (Magesh et al., 2012; Nasir et al., 2018; Martín-Loeches et al 2018),analytical hierarchy processes (Mohammadi-Behzad et al., 2019; Al-Djazouli et al., 2021), and Dempster-Shafer models (Mogaji and Lim 2018, Obeidavi et al 2021). Other frequently used expert methods are weight of evidence and frequency ratio analysis (Falah and Zeinivand, 2019; Boughariou et al., 2021). Machine learning is comparatively newer. A major difference between machine learning and expert approaches is that machine learning classification uses the advantages of artificial intelligence to find intricate associations among explanatory variables that might otherwise pass unnoticed. Hence, machine learning is well suited to map complex spatially-distributed variables such as groundwater occurrence. The GPM literature showcases a wide variety of supervised classification approaches. For instance,Al-Fugara et al. (2020) used mixed discriminant analysis to map spring potential in a watershed of Jordan, much like Odzemir (2011) mapped spring potential in a Turkish basin by means of a logistic regression method. Random forests have proved adept at mapping groundwater potential, both in mountain bedrock aquifers (Moghaddam et al., 2020) and large metasedimentary basins (Martínez-Santos and Renard, 2020). Other supervised classification methods used in the field of GPM include boosted regression trees (Naghibi et al., 2016), support vector machines (Naghibi et al., 2017b), neural networks

(Lee et al., 2012; Panahi et al., 2020) and ensemble methods (Naghibi et al., 2017a; Martínez-Santos and Renard, 2020; Nguyen et al., 2020b).

GPM is founded on the assumption that groundwater occurrence can be partially inferred from surface features. Some of the most frequently used explanatory variables in GPM studies include lithology, geological lineaments, landforms, topography,

soil, land use/land cover, drainage and slope-related variables, rainfall, and vegetation indices (Jha et al., 2007). Supervised classification algorithms are trained to find the associations between these variables and known groundwater data. Once the algorithms yield accurate predictions, their findings are extrapolated to estimate groundwater potential across a given study area.

The majority of machine learning GPM studies face two major shortcomings. Because the number of available boreholes to

train and test the algorithms is usually "small", and because the number of explanatory variables can be comparatively high, a crucial issue is how to reclassify explanatory variables to minimize noise and decrease the variability of each conditioning factor. Ultimately, categorical and continuous variables need to be reclassified either as integers or in intervals. Since reclassification relies heavily on the operator, this implies that bias may be incorporated from the beginning (Martínez-Santos and Renard, 2020).


A second problem is that the outcomes of machine learning studies are almost invariably assessed by means of standard big data metrics such as precision, recall, and area under the receiver operating characteristic curve (Pradhan, 2013; Naghibi et al., 2016; Chen et al., 2019). While useful, these may be of limited value in cases where the input dataset consists solely of unambiguous examples. Furthermore, there are question marks as to whether these metrics are truly representative when

developing spatially-distributed estimates (Martínez-Santos et al., 2021a). In some instances, using ad hoc calibration elements such as complementary borehole information can contribute to a better interpretation of the results.

The objective of this research is to build on the existing literature by presenting two methodological additions to machine learning GPM. In the first place, we explore different scaling methods to avoid the pitfalls associated with the reclassification

of explanatory variables. More specifically, we compare several automated approaches to data scaling, including standardization, normalization, maximum absolute value and max-min optimization (Pedregosa et al 2011). The second novelty has to do with the way the outcomes of this research are evaluated. In this context, borehole flow rates are used as a means to complement standard machine learning metrics, thus providing additional robustness to predictions. Our method is described in Figure 1, and demonstrated through its application to map groundwater potential across two regions of Mali. The

geographical setting also represents an added value to the literature because there are considerably fewer machine learning studies in Africa than in other continents (Naghibi et al., 2017a; Chen et al., 2018; Panahi et al., 2020).

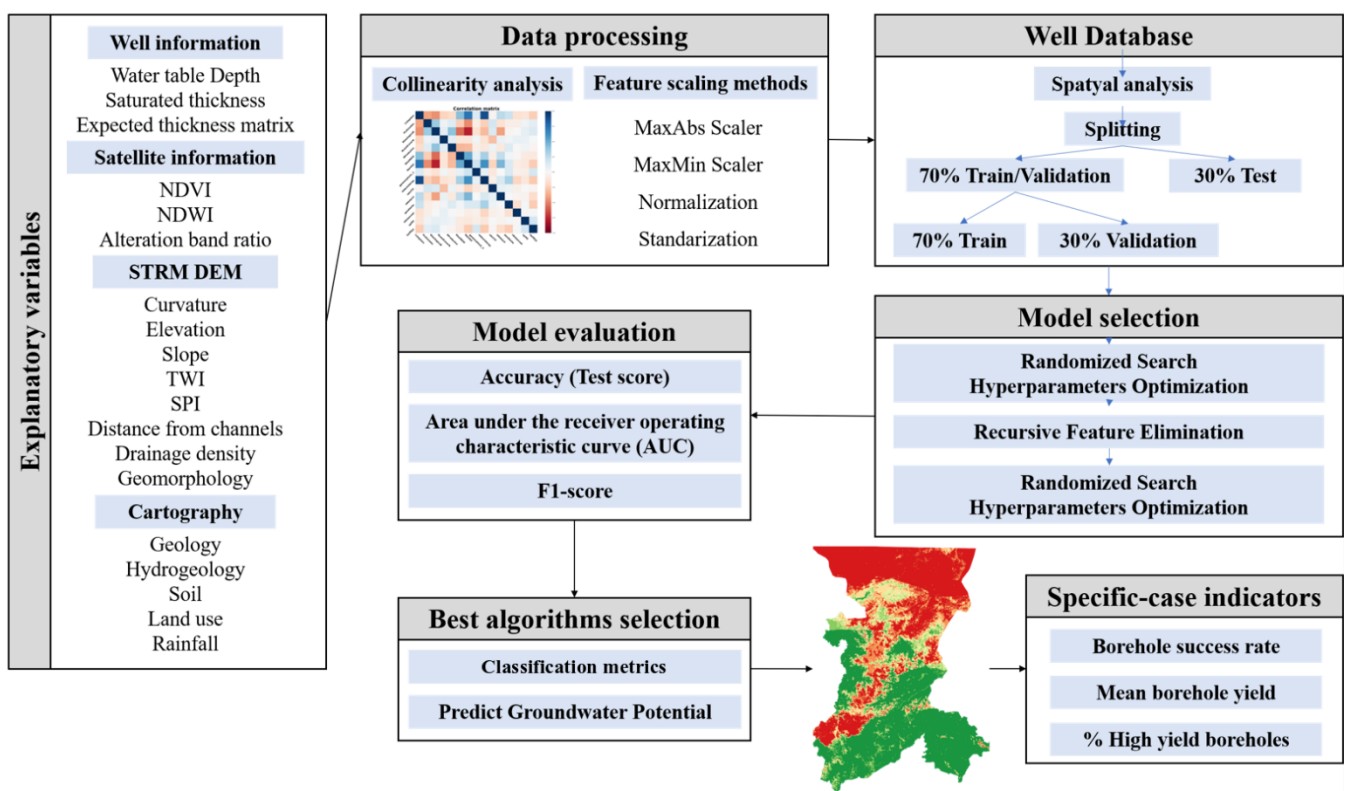

**Figure 1.** Conceptual model of the predictive mapping procedure with MLMapper v2.0.

## 2 Material and methods

### 2.1 Study area

This work pertains to the Koulikoro and Bamako regions of Mali (Fig. 2), which span a joint surface in excess of 90,000 km². The study area features three distinct climate zones, including tropical savannah in the south, hot arid steppe in the central and northern parts, and hot arid desert in the north (Traore et al., 2018). Mean yearly temperatures are relatively uniform from a spatial perspective (27 °C), although seasonal oscillations are observed. The coolest temperatures take place in the south in the month of January (19 °C) and the hottest in the north in May (41 °C). There is a clear rainfall gradient, from 300 mm/year in the north to 1200 mm/year in the south.

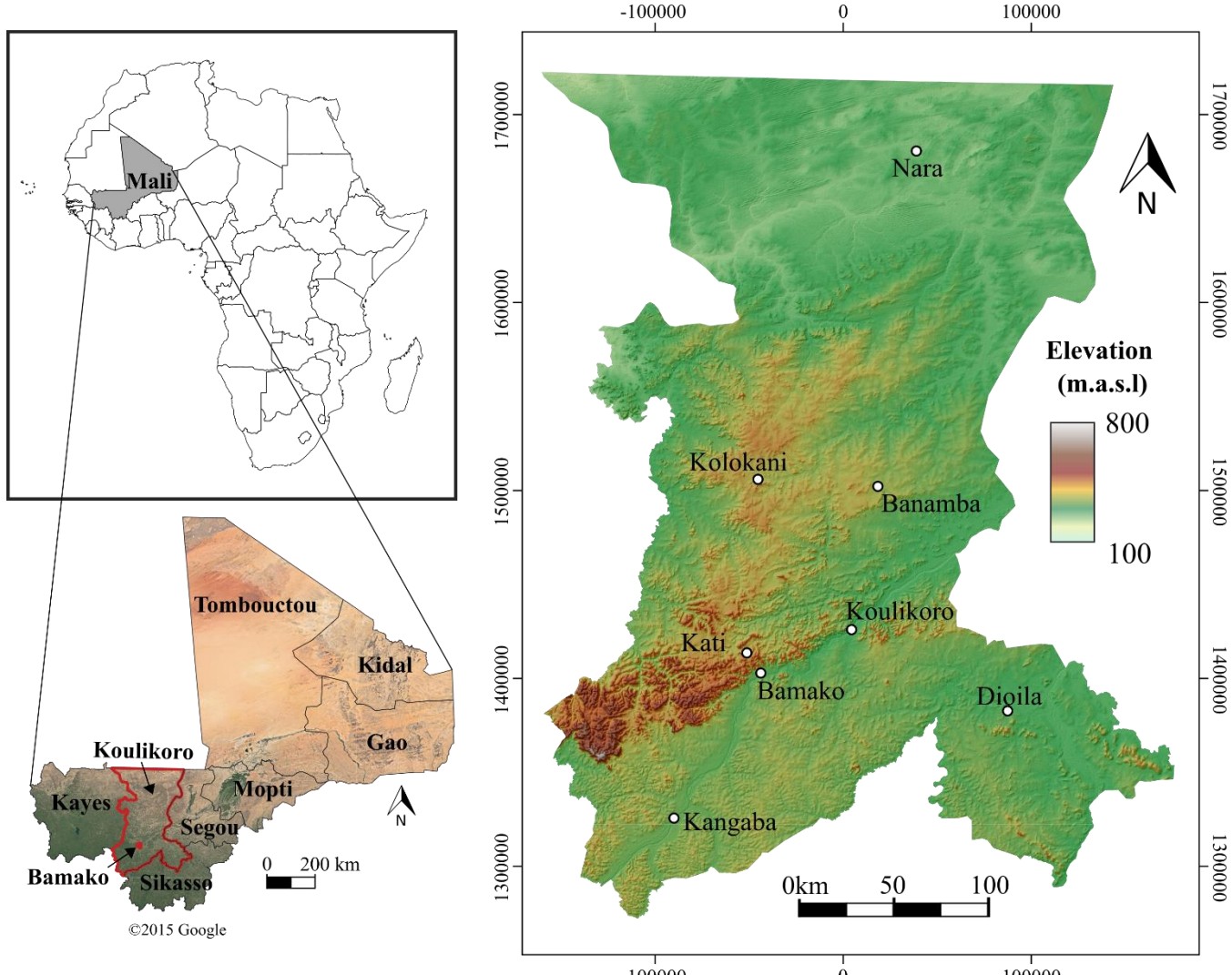

**Figure 2.** Study area. Koulikoro and Bamako regions located in southwestern Mali (from Map data ©2015 Google).

Figure 3 shows the major geological domains of the study area (BGS, 2021). The rocks that make up the Precambrian craton (south) are composed mainly of gneiss, schist and quartzite, representing metamorphosed volcanic-sedimentary sequences. The original sedimentary layers, which include shale, arkose, gravel and conglomerate, were intercalated with volcanic rocks, such as basalt, gabbro, dolerite, rhyolite and tuff. Further north, metasedimentary rocks of the Proterozoic age, predominantly low-medium grade metamorphosed sandstones, with varying amounts of mudstone and limestone, take up over 50% of the study area. Volcanic outcrops (basalts and gabbros) are located in the central sector and in the northern end. Sedimentary rocks (sandstone, limestone and shale) of the Cambrian-Carboniferous age and Cretaceous-Tertiary age occur in the northern third of the study area. Quaternary fluvial deposits associated with the Niger River are observed along the riverbed (Traore et al., 2018).

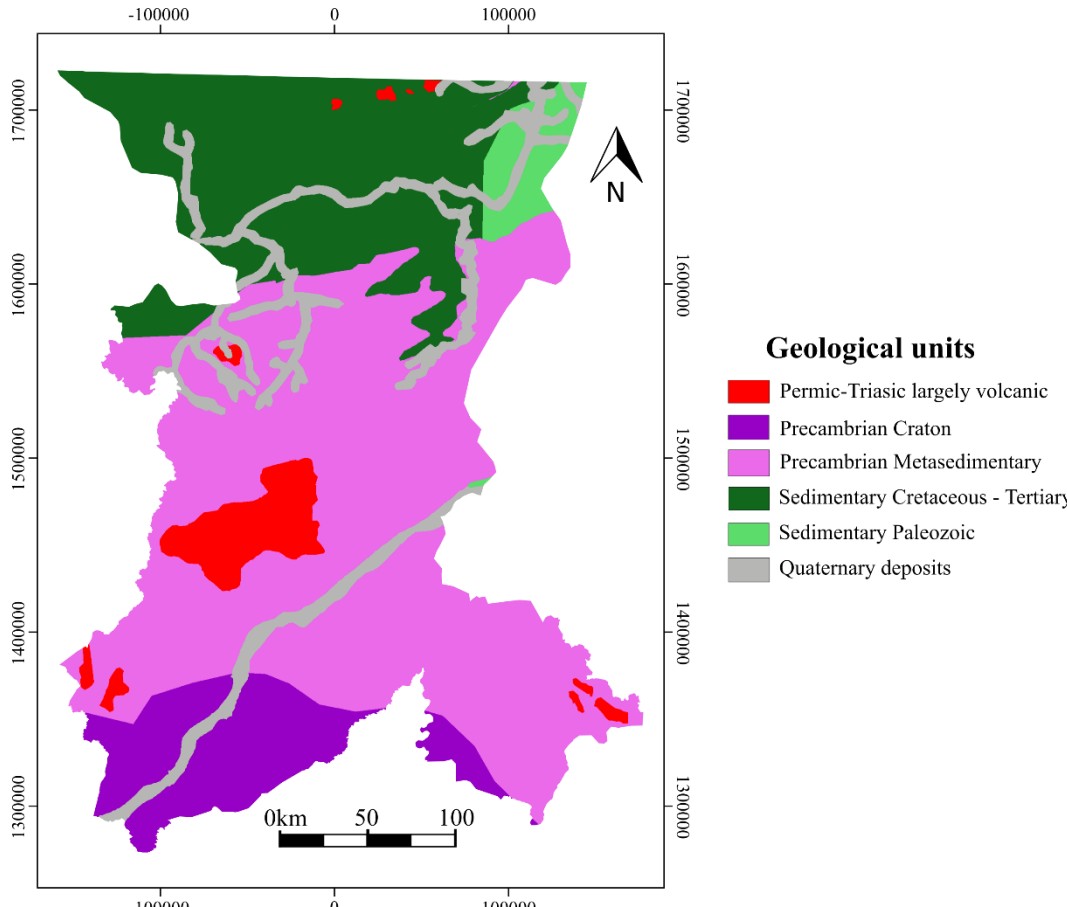

**Figure 3.** Geological map with the main units with outcrops in the study area (adapted from BGS, 2021)

From a hydrogeological perspective, four major aquifer units can be distinguished (Traore et al., 2018). These include basement aquifers, aquifers linked to fractures and intergranular porosity of consolidated sedimentary rocks (Precambrian and Paleozoic), aquifers formed in intrusive volcanic rocks, and aquifers in unconsolidated sedimentary materials (Fig. 4).

Basement aquifers are mostly located towards the south of the Koulikoro region. These are characterized by a thick weathered mantle. The average thickness of the weathered formation over the basement in this region is 10 to 50 metres. Groundwater flows preferentially through the weathered mantle. Within this, the lower part is generally more transmissive due to lower clay content. The upper part is less permeable to flow but can still be important as a groundwater reservoir. Fractures increase reservoir permeability although their storage capacity is typically low (Martín-Loeches et al., 2018). Borehole yields range 130 from 4 to 6 $m^3$/hour (Traore et al., 2018).

The Precambrian metasedimentary materials are located in the central part of the Koulikoro region. Metasediments are considered a mixed permeability aquifer: low permeability layers provide higher storage, while more fractured layers present

higher permeability and lower storage. Mean aquifer thickness ranges from 30 to 50 metres and the average yield varies from
5 to 10 m³/hour. However, some boreholes exceed 100 m³/hour. In the north, the fractured Paleozoic rocks allow water to flow
through the sandstone and limestone layers. Average borehole yields are around 6 m³/hour and the fractured horizons are about
40-45 m thick. Finally, unconsolidated sedimentary materials are composed of shales and argillaceous sandstone interbedded
with limestone. The average borehole yields around 7 m³/hour. The thickness of the saturated zone ranges from less than 100
m to over 400 m (Traore et al., 2018).

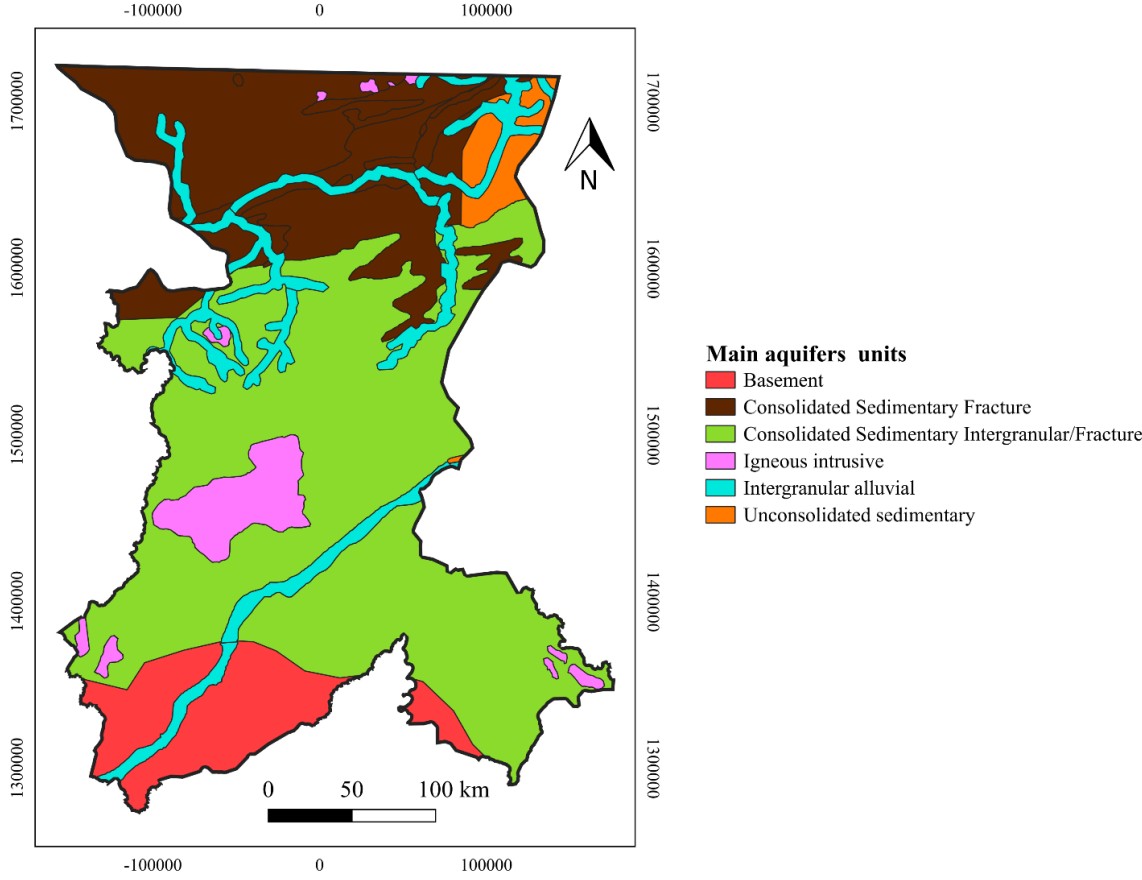


**Figure 4.** Main aquifer units of the study area (Adapted from Traore et al. (2018))

## 2.2 Data

Borehole data were provided by the National Water Directorate of Mali (DNH, 2010). The database contains information on
5,387 boreholes (3,772 successful and 1,615 unsuccessful), distributed across 1,605 human settlements. The smallest unit of
data aggregation is the village level. Available information includes the total number of boreholes per settlement, the borehole
success rate, the mean flow rate (m³/h), and the number of boreholes within a given flow rate range (less than 5 m³/h, between
5-10 m³/h and over 10 m³/h). In most cases, there is also information on the mean borehole depth (m), mean static water table
depth (m) and mean electric conductivity (µS/cm). Figure 5 shows that the most common yields range between 1.5 and 4.5

m³/h, while the mean borehole depth ranges between 50-80 metres. This can be assumed to be the thickness of the weathered

formation over the fresh granite and is roughly consistent with studies conducted in Burkina Faso (Courtois et al., 2010). Water

table depth ranges mostly between 5 and 15 metres.

There is a considerable number of villages with a 100% success rate (530), many of which have a single borehole (452). This

raises the question of  whether villages with a small number of boreholes are statistically representative, particularly in cases

where the mean yield is low. Besides, it creates an imbalance in the input dataset. Villages with fewer than five boreholes were

therefore omitted from the sample.

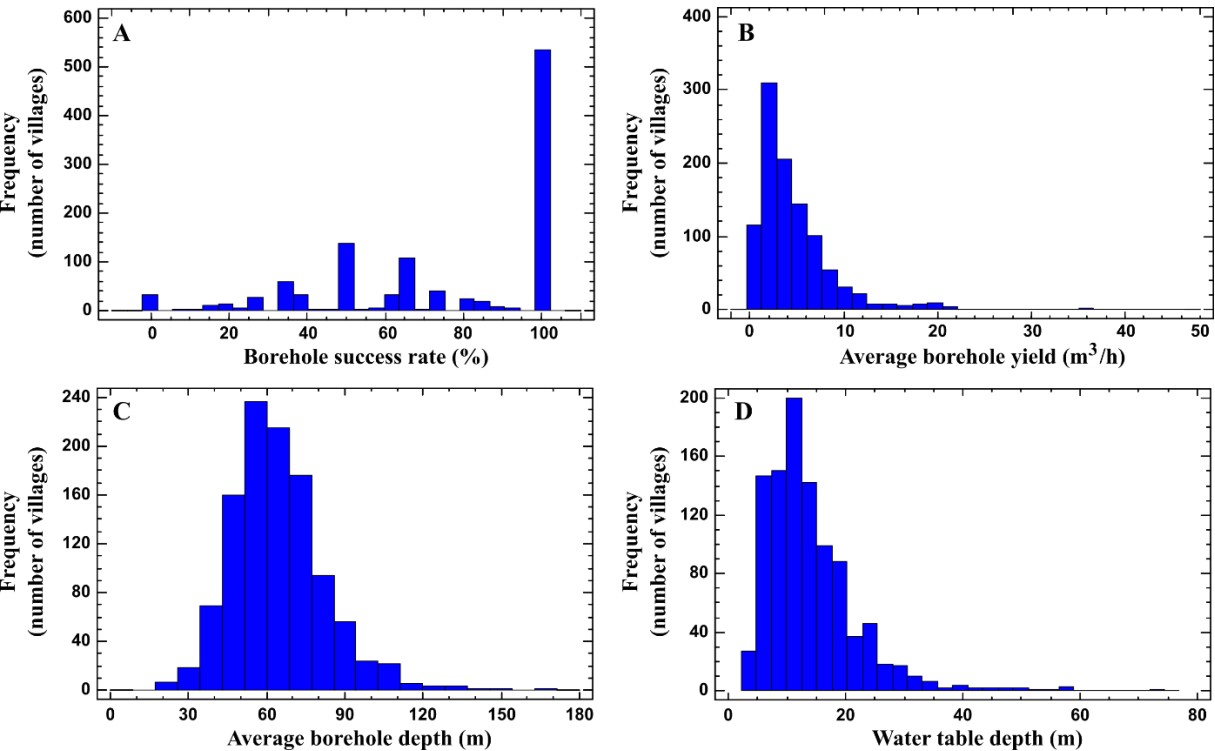

**Figure 5.** Frequency analysis of the four borehole-related variables. (A) Borehole success rate (B) Average borehole yield (C) Average borehole depth and (D) Water table depth

**2.2.1 Target variable**

For the purpose of algorithm training, groundwater potential is defined as the likelihood of a drilled borehole being successful.

Successful boreholes are those that yield sufficient water to justify the installation of a hand pump (>0.5 m3/h) (Foster et al.,

2006). The target variable is therefore binary, and can be interpreted as the presence/absence of groundwater.

Villages where more than 50% of wells were known to be successful were labelled "positive". The positive classification also

applies to those villages with more than one high yield borehole (>10m3/h). The others were labelled "negative". The resulting

input dataset consisted of 650 villages, out of which 390 were labelled positive and 260 were negative. This comprises information from 3,345 boreholes, out of which 2,101 were successful and 1,244 were unsuccessful.

### 2.2.2 Explanatory variables

Groundwater potential depends largely on groundwater recharge. In turn, recharge is influenced by five main factors (Kumar, 1997; Jyrkama et al., 2002), namely climate (e.g. precipitation, temperature, potential evapotranspiration), soils (e.g. texture, saturated hydraulic conductivity, moisture capacity), land cover (e.g. vegetation density and type), geomorphology (e.g. surface slope, drainage density) and hydrology (e.g. streamflow, water table depth).

Rainfall is the principal source of groundwater recharge. Recharge depends on precipitation rate, as well as on the surface and subsurface factors that will allow or prevent infiltration. Soil is important because its characteristics (permeability, grain shape, grain size, and void ratio) control percolation. Higher infiltration potential is associated with sandy and gravelly soils, while clayey and silty soils rank among the least favorable for recharge (Díaz-Alcaide and Martínez-Santos, 2019). Integration of land use and land cover is often used in groundwater potential mapping studies because land use changes, which are mostly induced by human activities, affect hydrological dynamics (Díaz-Alcaide and Martínez-Santos, 2019). For instance, croplands and forests, located in the southern part of the study area, could be associated with high groundwater potential because ploughing, root development and biological activity favour infiltration. Areas close to permanent water bodies also tend to correlate with a higher groundwater potential (Naghibi et al., 2017a). In contrast, urban settlements and wastelands are assumed to have low groundwater potential due to the presence of impervious surfaces, as well as to the absence of moisture (Magesh et al., 2012). Geomorphology may be useful in identifying features that may be favorable for groundwater infiltration and storage (Díaz-Alcaide and Martínez-Santos, 2019). Alluvial fans, sand dunes, weathering mantles, floodplains, and other accumulations of unconsolidated materials are generally recognized as the most interesting geomorphological features from a groundwater point of view (Venkateswaran and Ayyandurai, 2015). In contrast, landforms such as inselbergs, scarps, and ridges may be considered of little interest.

Hydrological factors such as drainage density or water table depth also play an important role in groundwater recharge. High drainage density means that runoff can be evacuated quickly and therefore infiltration is less likely. (Magesh et al. 2012; Fashae et al. 2014). In addition, a high drainage density can be assimilated to a higher erosion potential. Meijerink (2007) shows that parallels can be found between drainage density and soil permeability in certain geological settings. Water table depth is useful for mapping water tables to determine the main recharge and discharge zones of an aquifer.

Nineteen explanatory variables were selected from an extensive review of the literature on GPM (Diaz-Alcaide and Martinez-Santos, 2019). Explanatory variables (Table 1) include lithology (Fig. 2), landforms, land use, soil, expected thickness (Fig. 5), rainfall, water table depth, vegetation-related indices (NDVI, NDWI), slope curvature, slope, topographic wetness index,

stream power index, drainage density, distance from channels, clay content and clay mineral alteration ratio (Fig. 6). An additional layer with mean borehole flow rates per village was developed for the purpose of calibrating the results.

**Table 1**. Explanatory variables used in GPM. The scale/resolution, acquisition time and source of data for each factor are provided.

| Explanatory variables | Scale/resolution | Time (dd/mm/yyyy) | Source of data |
|---|---|---|---|
| Alteration Band Ratio | 30 metres | 07-16/03/2020 | Own elaboration from Landsat 8 |
| Clay content | 250 metres | N/A | SoilGrids250m 2.0 |
| Curvature | 30.53 metres | N/A | Own elaboration from DEM |
| Saturated thickness | 30.53 metres | N/A | Own elaboration from DEM and borehole database |
| Water table Depth | 30 metres | 2010 | Own elaboration from Borehole database |
| Distance from channels | 30.53 metres | N/A | Own elaboration from DEM |
| Geology | 1:5 million | N/A | British Geological Survey |
| Geomorphology | 30.53 metres | N/A | Own elaboration from DEM |
| Land use | 300 metres | 2009 | ESA Climate Change Initiative |
| Soil | 1:3M | N/A | European Soil Data Centre |
| Rainfall | 0.5° | 1950-2009 | CRU TS 3.21 dataset (Climatic Research Unit at the University of East Anglia) |
| Drainaige density | 30.53 metres | N/A | Own elaboration from DEM |
| Thickness matrix | 30.53 metres | N/A | Derived from DEM and borehole database |
| Elevation (DEM) | 30.53 metres | 23/09/2014 | Shuttle Radar Topography Mission (SRTM) |
| NDVI | 30 metres | 07-16/03/2020 | Own elaboration from Landsat 8 |
| NDWI | 30 metres | 07-16/03/2020 | Own elaboration from Landsat 8 |
| Slope | 30.53 metres | N/A | Own elaboration from DEM |
| SPI | 30.53 metres | N/A | Own elaboration from DEM |
| TWI | 30.53 metres | N/A | Own elaboration from DEM |

QGIS 3.0's Geomorphon plugin (Jasiewicz and Stepinski, 2013) was used to prepare the landform map. This approach uses DEM for the classification and mapping of landform features based on the principle of pattern recognition, rather than on differential geometry. By default the Geomorphon plugin classifies landforms into ten different categories. Because some of

them can be expected to play a similar role in the context of GPM, these were subsequently regrouped into four (Fig. 6a). These comprise flat areas, gentle slopes, steep slopes and ridges.


Soil descriptions (Fig. 6b) of the study area were obtained from the European Soil Data Centre (Dewitte et al., 2013). About 45% of the region is characterized by the presence of Pisoplinthic Plinthosols a type of soils with plinthite (Fe-rich), strongly cemented to indurated concretions or nodules, humus-poor mixture of kaolinitic clay and other products of strong weathering (IUSS Working Group, 2015). It usually changes irreversibly to a layer with hard concretions or nodules or to a hardpan on

exposure to repeated wetting and drying. Hypoluvic arenosols are present in 20% of the total surface and characterized by being deep sandy soils, which explains their generally high permeability. These are residual sandy soils following weathering of rocks, generally rich in quartz, in situ. Nearly 13% of the study is characterized by Petric Plinthosols, which share multiple features with Pisoplinthic Plinthosols but which, unlike the latter, are arranged into continuous or fractured sheets of connected concretions or nodules and are strongly cemented to indurated layer starting layer starting $\leq 100$ cm from the soil surface.

Lithic Leptosols (very thin soils on continuous rock extremely rich in coarse fragments with continuous rock from $\leq 10$ cm from the soil surface), Haplic Lixisols (higher clay content in the subsoil than in the topsoil, as a result of pedogenetic processes), Eutric Regosols (very weakly developed mineral soils in unconsolidated materials) constitute about 15% of the study area.

Land cover seems to present a clear association with the precipitation gradient (Fig. 6c). The southern part is characterized by open broadleaf deciduous forests (ESA, 2010). The central part is characterized by an alternation of shrublands, mosaics of cropland vegetation and rainfed cropland. West of Bamako, in the sparsely populated mountains, there are forests mixed with shrublands. The northern part of the study areais dominated by cropland mosaics and, further north, the landscape is made up of open grasslands, sparse vegetation and bare areas.


Boreholes in the study area are often drilled until the unaltered bedrock is reached. As a result, borehole depth can be a suitable proxy for aquifer thickness (Fig. 6d). Because the borehole database includes static level measurements, an expected saturated thickness layer was computed by subtracting one from the other (Fig. 6d).

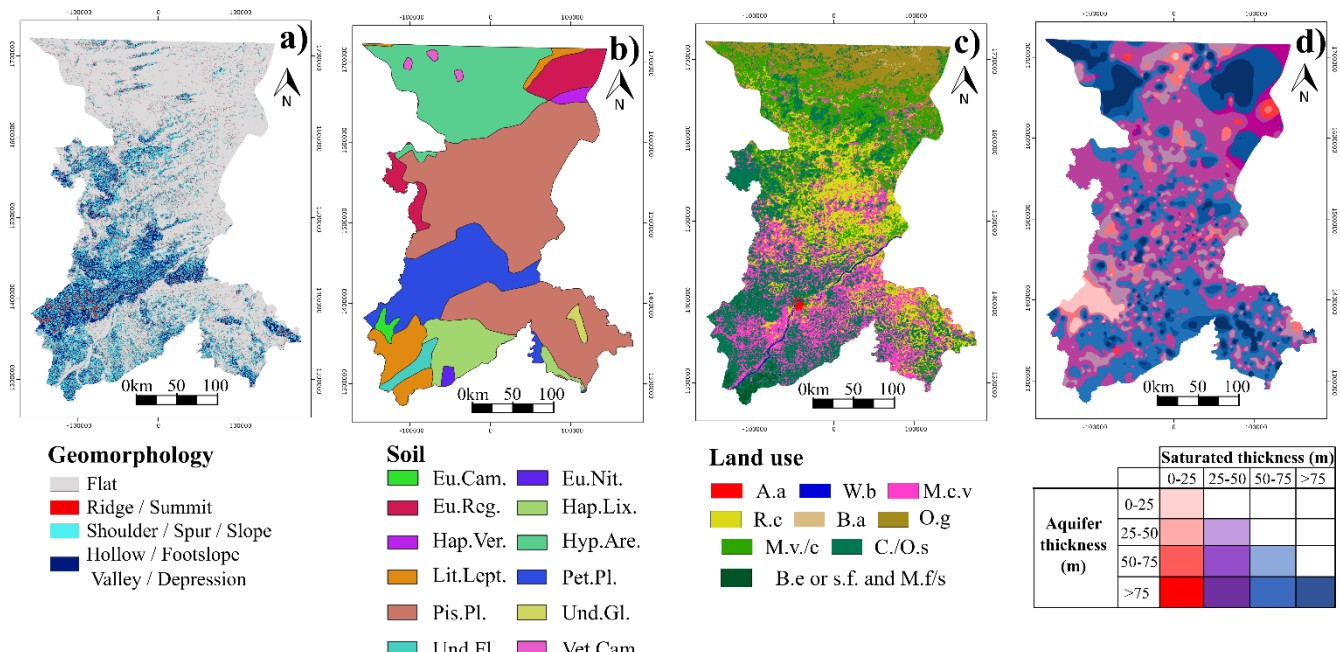

**Figure 6.** Explanatory variables used to predict the GPM: a) geomorphology b) Land use (A.a = artificial areas; W.b = water bodies; M.c.v = Mosaic cropland vegetation; R,c = Rainfed cropland; B.a = Bare areas; O.g = Open grassland; M.v./c = Mosaic vegetation/cropland; C./O.s = Close to open shrubland; B.e or s.f. and M.f/s = Broadleaved evergreen or semidecidous forest and Mosaic forest / shrubland) c) Soil (Eu.Cam. = Eutric Cambisols; Eu.Nit. = Eutric Nitrisols; Eu.Reg. = Eutric Regosols; Hap.Lix. = Haplic Lixisols; Hap.Ver. = Haplic Vertisols; Hyp.Are. = Hypoluvic Arenosols; Lit.Lept. = Lithic Leptosols; Pet.Pl. =Petric Plinthosols; Pis.Pl. = Pisoplinthic Plithosols; Und.Gl. = Undifferentiated Gleysols; Und.Fl. = Undifferentiated Fluvisols; Vet.Cam = Vetric Cambisols) d) expected thickness matrix.

Satellite monitoring does not penetrate deep into the ground, but provides information about features that may be associated with shallow groundwater (Díaz-Alcaide and Martínez-Santos, 2019). This can be important in the case at hand, where the borehole database shows the static level to remain around 5-15 m below the surface (Fig. 7b). Vegetation-related indices can be useful in this context, particularly when computed at the end of the dry season (Fig. 7c,d). Take for instance the normalized difference vegetation index (NDVI, Fig. 7c), which is an estimate of vegetation vigour. This index is derived from the response of vegetation to red and visible infrared wavelengths (Xie et al., 2008). Similarly, the normalized difference water index (NDWI, Fig. 7d) is used as a measure of the amount of water in the vegetation or soil moisture (Xu, 2006). Based on Landsat 8 products, the NDVI and the NDWI are computed as per Eq. 1 and Eq. 2, respectively, where B3 represents the green band (0.53 - 0.59 μm), B4 is the red band (0.64 - 0.67 μm) and B5 is the near infrared band (0.85 - 0.88 μm).

$$\text{NDVI (Landsat 8)} = (B5 - B4) / (B5 + B4) \tag{1}$$
$$\text{NDWI (Landsat 8)} = (B3 - B5) / (B3 + B5) \tag{2}$$

The digital elevation model (DEM) was obtained from the radar-based Shuttle Radar Topography Mission (SRTM), with a resolution of 1 arcsecond (30 m). Topography is a relevant factor in groundwater distribution, storage, and flow. Surface runoff and infiltration are also constrained by surface features (Elbeih, 2015). In this case, the DEM was used to develop the curvature (Fig. 7e), slope (Fig. 7f), topographic wetness index (TWI, Fig. 7g) and stream power index (SPI, Fig. 7h). The Topographic Wetness Index (TWI) represents the ease with which water may accumulate at the surface (Beven and Kirkby, 1979). Similarly, the Stream Power Index provides a measure of the erosive power of flowing water (Moore et al., 1991). Other features computed from topographic data include the landforms (Fig. 6a) and drainage-related layers (Fig. 7i and 7j).

The channels extracted from the DEM are used to develop the drainage density and distance from channels maps. Drainage density is computed as the total length of the streams per catchment unit area. Distance from channels was developed by extracting all major channels into a separate layer and developing 500-, 1500- and 2500-metre buffers.

Clay content in the first few metres of the surface largely determines the percolation of water into the aquifer. Therefore, an additional clay content layer (g/kg) in the top two metres of the terrain was considered (Poggio and de Sousa, 2020). This layer is obtained by state-of-the-art machine learning methods that use global soil profile information and covariate data to model the spatial distribution of soil properties around the world. (Fig. 7k). This layer provides information about subsurface clay content. To complement the information on clay content on the subsurface, an additional layer has been developed by combining bands 6 (short-wave infrared 1) and 7 (short-wave infrared 2) of Landsat 8 (Ourhzif et al., 2019). This layer provides information on clay content on the surface and the relationship with infiltration (Fig. 7l).

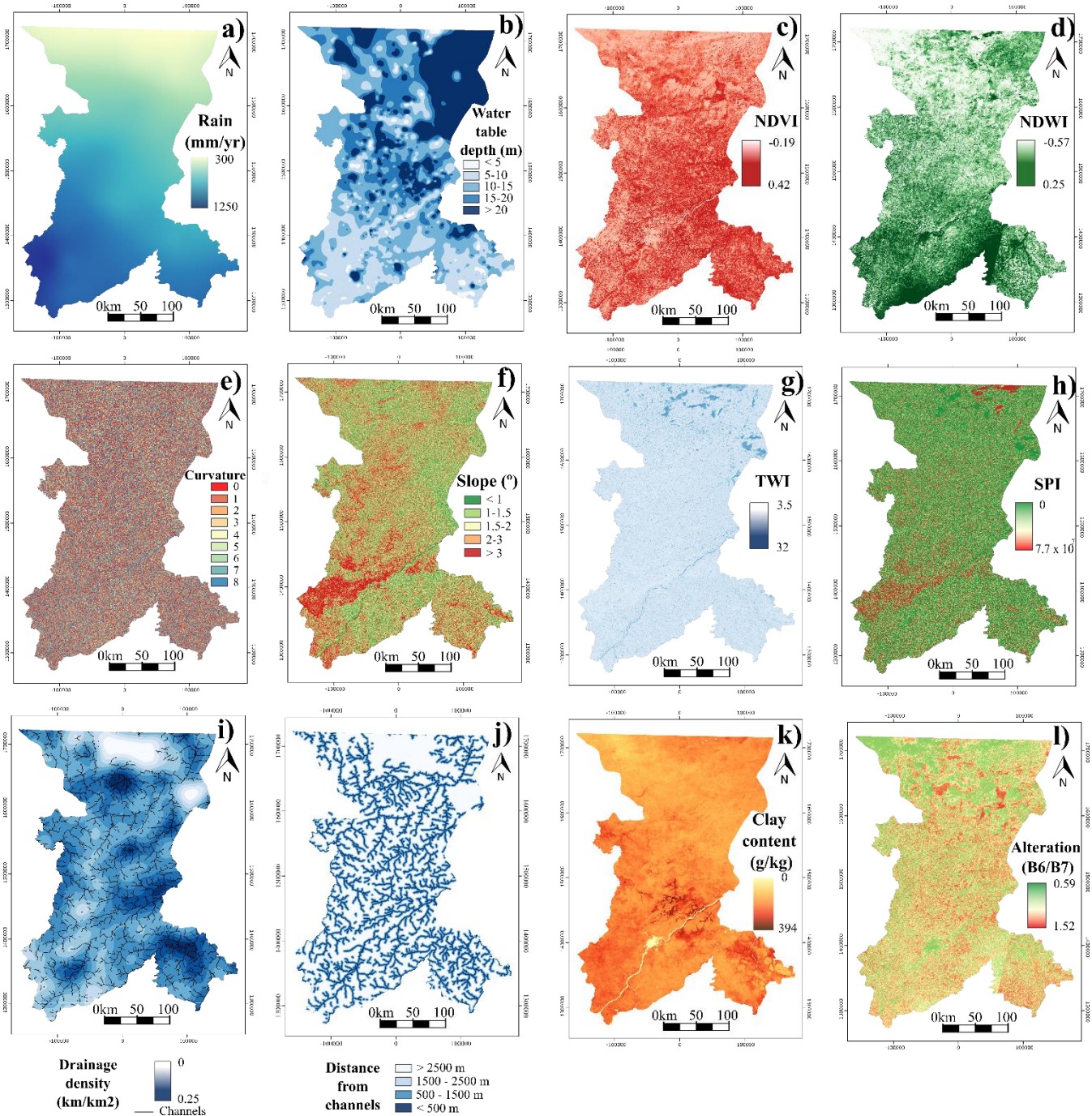

275

**Figure 7.** Explanatory variables used to predict the GPM: a) rainfall (mm/year) b) water table depth (metres) normalized difference vegetation index (NDVI) e) curvature f) slope (degree) g) topographic wetness index (TWI) h) stream power index (SPI) i) drainage density j) distance from channels k) clay content (g/kg) l) alteration band ratio (B6/B7)

## 2.3 Predictive mapping software

MLMapper v2.0 is an upgrade of the software developed by Martínez-Santos and Renard (2020). MLMapper is a QGIS3 plugin that allows for the development of predictive maps of a given target variable based on a series of explanatory variables (Fig. 1). It relies on a variety of supervised learning classifiers from the SciKit-Learn 0.24.1 toolbox (Pedregosa et al., 2011). Version 2.0 uses 19 different algorithms, including support vector machines (SVC), linear support vector machines (LVC), logistic regression (LRG), decision tree classifier (CRT), random forest classifier (RFC), K-neighbour classification (KNN), linear discriminant analysis (LDA), Gaussian naïve Bayes classification (NBA), multilayer perceptron neural network (MLP), Ada-boost classifier (ABC), quadratic discriminant analysis (QDA), gradient boosting classification (GBC), Gaussian process classifier (GPC), ridge classifier (RID), stochastic gradient descent linear classifier (SGD), perceptron (PER), passive aggressive classifier (PASSI), nu-support vector classifier (nuSVC), and extra-trees classifier (ETC).

## 2.4 Preprocessing of explanatory variables

Machine learning algorithms are designed to find meaningful associations between the explanatory variables and a given target variable. A preprocessing of explanatory variables is important to avoid bias. Take for instance the issue of ranges. Rainfall in the study area oscillates between 300 and 1200 mm/year, while NDVI and NDWI values only range between -1 and +1. Since some algorithms attribute weight to explanatory variables based on sheer magnitude, scaling is advisable to prevent the rainfall layer from having an undue bearing on the results. Furthermore, those algorithms that use gradient descent as an optimization technique (logistic regression, neural networks), require data to be scaled because range differences lead to different step sizes in the gradient descent formula for each feature, and computational time increases as a result. Algorithms such as K-neighbors and support vector machines are affected by different ranges because these classifiers rely on distances between data points to determine their similarity.

Scaling is advocated as an essential part of algorithm training, since each subsequent procedure depends on the choice of unit for each feature (Huang et al., 2015). Furthermore, scaling is expected to transform feature values based on a defined rule, so that all scaled features have the same degree of influence (Angelis and Stamelos, 2000).

In this context, the rationale behind using preprocessing approaches is to rely on raw data as much as possible, instead of reclassifying it into intervals generated statistically or by expert criteria. Four scaling methods were therefore used: standardization, maximal absolute scaler (MaxAbs), maximal-minimal scaler (MaxMin) and normalization (Pedregosa et al., 2011).

The standardized explanatory variables are obtained by removing the mean and scaling to unit variance (Zheng and Casari, 2018). Centring and scaling happen independently on each explanatory variable by computing the relevant statistics on the

samples in the training set. Mean and standard deviation are then stored to be used on later data using transform. This is obtained by means of Eq. 3, where $\tilde{x}$ is the new value of the sample, x is the old value of the sample, u is the mean of the training samples, and s is the standard deviation of the training samples.

$\tilde{x} = (x - u) / s$                                                                     (3)

The MinMax method transforms features by scaling each explanatory variable to a given range. This estimator scales and translates each feature individually so that it lies within the range given in the training set (between zero and one in the case at hand). Thus, this scaling method is robust to small standard deviations in feature data (Pedregosa et al., 2011). MinMax scaling is performed as per Eq.4.

$\tilde{x} = (x - min(x))/(max(x) - min(x))$                                                  (4)

The MaxAbsScaler works in a similar way, but the training data is scaled to fall within the [-1, 1] range by dividing by the largest maximum value of each feature. Finally, under the normalizer procedure, each sample is rescaled regardless of other samples so that its norm ('L1', 'L2' or 'max') equals one (Pedregosa et al., 2011). The L1 norm attempts to minimize the sum
of the absolute differences between the target value and the estimated values. In turn, the L2 norm uses least squares, while the L-Max norm rescales by the maximum of the absolute values. The L2 norm is by far the most common and was adopted as the procedure of choice.

### 2.5 Supervised classification routine

MLMapper incorporates a series of routines to enhance the conventional train/test process (Fig. 1). These include collinearity checks, random-search parameter fitting, and recursive feature elimination. A selection of the best algorithms and a basic arithmetic ensemble is also performed at the end in order to appraise the degree of agreement among classifiers.

Collinearity occurs when two or more variables are highly correlated. This can affect the performance of the classifiers by
attributing extra weight to an input variable or by adding noise to the final outcomes. Interpretability can also be impaired because the regression coefficients of certain algorithms are not uniquely determined (Martínez-Santos et al., 2021a). MLMapper incorporates a collinearity analysis function to prevent collinearity from adversely affecting the results. Collinearity analysis is performed before running the algorithms. Pairwise correlation among explanatory variables is computed and correlation coefficients are expressed in a range between -1.0 (inverse correlation) and 1.0 (direct correlation).
Highly-correlated explanatory variables may be excluded from the algorithm training procedure

Random-search parameter fitting increases the accuracy of the predictions by identifying the best combination of those parameters that govern each algorithm. The random search cross-validation function needs an algorithm, a scoring metric to evaluate the performance of the different hyperparameters and a dictionary with the hyperparameter names and values. In regard to the previous version of MLMapper, which used grid-search cross-validation, this provides additional flexibility and reduces computational time. A sensitivity analysis of the number of iterations was performed. The best compromise between the results and computational cost was obtained by fixing the number of iterations at 500. No significant improvement in scoring metrics was observed for higher values, but running times increased considerably.

Once the optimal parameters distributions have been selected, recursive feature elimination cross-validation (RFECV) is performed. RFECV is only available for those algorithms that use feature importance or coefficient weight attributes. The goal of RFECV is to identify the most important explanatory variables for each algorithm, as well as to eliminate those features that either incorporate noise or detract from the accuracy of predictions.

Classifiers are trained first on the initial set of explanatory variables. The weight of each feature is obtained, either through a coefficient attribute or through a feature importance attribute. The least important explanatory variables are then pruned from the model. This procedure is recursively repeated on the pruned set until the optimal number of features to  be selected is eventually reached (Pedregosa et al., 2011). Once RFECV is complete, algorithms are run again on the input dataset (excluding noise and counterproductive variables) and hyperparameters are re-optimized. This step yields important information for the purpose of interpreting the maps, as it provides the user with the relative weight of explanatory variables.

Each algorithm predicts groundwater potential throughout the study area, assigning a value of 1 to pixels with positive groundwater potential and 0 to pixels where groundwater is expected to be absent. Each algorithm operates differently and relies on a different combination of explanatory variables, which inevitably leads to discrepancies in the predictions. In order to analyse the degree of agreement between the classifiers, an ensemble map is developed by computing the arithmetic mean at the pixel scale of those algorithms exceeding 0.85 predictive accuracy. Green pixels mean that all the best-performing algorithms agreed on a positive groundwater potential outcome (arithmetic mean = 1). Conversely, red zones represent those pixels where all the best-performing algorithms agreed on a negative groundwater potential (arithmetic mean = 0). Intermediate colours represent various degrees of agreement among the algorithms.

## 2.5 Machine learning metrics for algorithm evaluation

Outcomes of machine learning studies are often evaluated on the grounds of accuracy (test score) and the area under the receiver operating characteristic curve (AUC). The test score is calculated as the number of successful predictions over the total number of attempts in the test dataset, thus providing a direct measure of predictive accuracy. In turn, the receiver

operating characteristic curve shows the performance of a classification model at all classification thresholds. The AUC is a probabilistic metric that assesses the degree to which algorithms can distinguish between classes. Test score and AUC rank algorithms on a scale of 0 to 1, a higher score implying better performance. If the classes are balanced, a score of 0.5 on either metric suggests that the prediction is no better than a random estimate.

Because the ratio between positive and negative classes in the input dataset is approximately 3:2, the F-1 metric was also used to analyse the quality of the predictions in each of the classes separately. The F1-score is the harmonic mean of precision and recall. Precision is calculated like the ratio of true positives over the sum of true and false positives. Precision therefore shows the classifier's ability not to label as positive a sample that is negative. In turn, recall is the ratio of true positives over the sum of true positives and false negatives. In other words, recall represents the ability of the classifier to find all positive samples.

**3 Results and discussion**

**3.1 Collinearity analysis**

Figure 8 presents the results of the collinearity analysis. Pair-wise correlation coefficients are expressed as a colour palette ranging from -1.0 (inverse correlation) to 1.0 (direct correlation). There is no hard threshold as to what is an acceptable level of correlation between two variables, though the literature shows that values between 0.4 and 0.85 may be acceptable (Dormann
et al., 2013). In this case, no strong correlations were found among explanatory variables except when appraising the expected saturated thickness versus the expected thickness matrix. This pair renders a correlation coefficient of 0.85, which is close to the upper acceptability threshold. Because the number of non-surface features in the explanatory dataset is limited, a closer look was taken at the importance of these two variables. RFECV reveals that both are relied upon by the best performing algorithms. In addition, the thickness matrix has seldom been used in the literature. These reasons led to the decision of keeping
both variables.

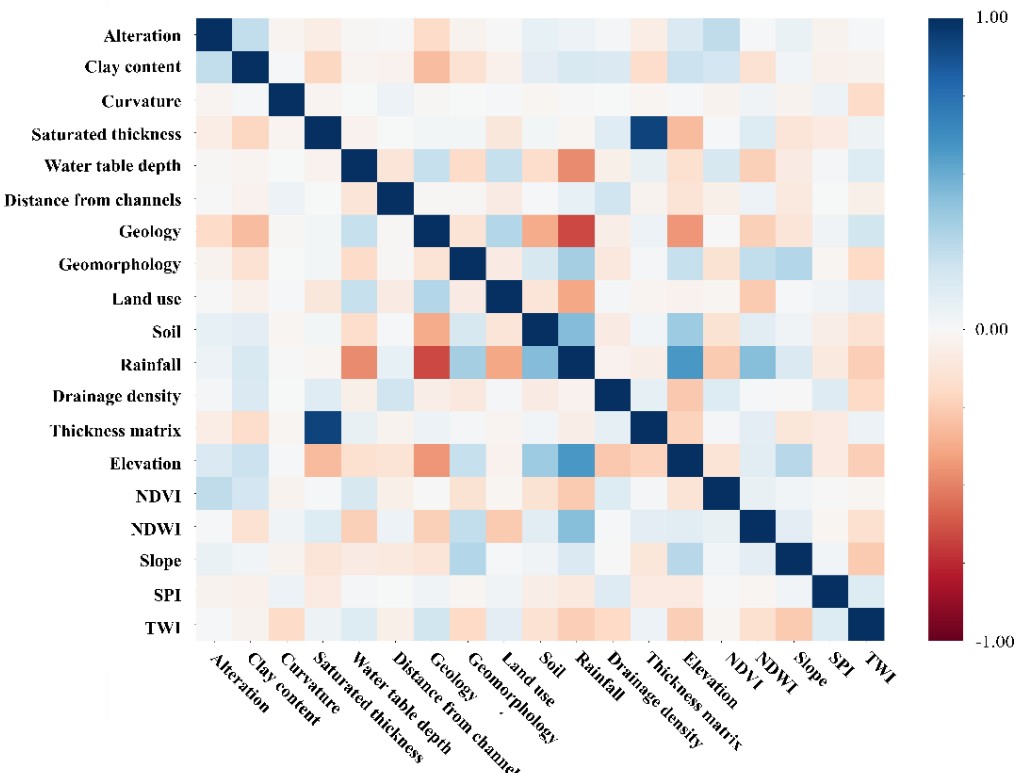

**Figure 8.** Pair-wise collinearity analysis for all explanatory variables.

### 3.2 Model evaluation and scaling method selection

Table 2 shows the results of the best algorithms for each scaling method. The five best algorithms for each scaling method
were selected and the mean of each metric was computed to analyse performance. All scaling methods render a mean test score
for the top five algorithms in excess of 0.85, which implies that the predictions are accurate in all cases. The standardization
scaling method presents the highest value (0.867), followed by the MaxAbs method (0.866), the normalization method (0.860)
and the MaxMin method (0.857). All algorithms render a higher F1 score for the positive outcome, which is the most common
one in the input database. However, the F1 score for the negative class is always higher than 0.81. This indicates that the
algorithms are capable of distinguishing positive and negative classes accurately.

AUC exceeds 0.90 in all cases except for the AdaBoost algorithm (0.898) and Decision Tree algorithm (0.893). This implies
that all the best-performing algorithms have a high probability of adequately predicting the target variable. When comparing
the scaling methods, the standardized one presents the highest scores for test score and F1 score for the negative class while

the MaxAbs method present the second-best performance in terms of test scoring, F1 score for the negative class and AUC

metrics. Hyperparameter optimization by the random search procedure led to the scores in Table 3.

**Table 2.** Results of the top five algorithms for each scaling method. Refer to section 2.6 for the definition of each metric (Test score = optimized test score; F1 score 0 = f-1 score false; F1 score 1 = f-1 score true; AUC = area under curve). The mean of each metric has been

calculated for the different reclassification methods to compare the results.

| Scaling method | Better algorihms | Test Score | Test score mean | F1 score 0 | F1 score 0 mean | F1 score 1 | F1 score 1 mean | AUC | AUC mean |
|---|---|---|---|---|---|---|---|---|---|
| Standarize | Decision Tree | 0.872 | | 0.850 | | 0.890 | | 0.893 | |
| | Random Forest | 0.852 | | 0.820 | | 0.870 | | 0.911 | |
| | AdaBoost | 0.862 | 0.867 | 0.830 | 0.836 | 0.880 | 0.886 | 0.912 | 0.909 |
| | Gradient Boosting | 0.872 | | 0.840 | | 0.890 | | 0.914 | |
| | Extra Trees | 0.878 | | 0.840 | | 0.900 | | 0.914 | |
| Normalization | RandomForestClassifier | 0.847 | | 0.810 | | 0.870 | | 0.901 | |
| | GradientBoostingClassifier | 0.857 | | 0.830 | | 0.880 | | 0.905 | |
| | SGDClassifier | 0.857 | 0.860 | 0.830 | 0.828 | 0.880 | 0.882 | - | 0.907 |
| | PassiveAggressiveClassifier | 0.862 | | 0.830 | | 0.880 | | - | |
| | ExtraTreesClassifier | 0.878 | | 0.840 | | 0.900 | | 0.916 | |
| MaxMin | Decision Tree | 0.857 | | 0.820 | | 0.880 | | 0.913 | |
| | Random Forest | 0.857 | | 0.830 | | 0.880 | | 0.912 | |
| | Gradient Boosting | 0.847 | 0.857 | 0.820 | 0.824 | 0.870 | 0.880 | 0.903 | 0.911 |
| | Ridge Classifier | 0.842 | | 0.800 | | 0.870 | | - | |
| | Extra Trees | 0.883 | | 0.850 | | 0.900 | | 0.918 | |
| MaxAbs | Decision Tree | 0.867 | | 0.840 | | 0.890 | | 0.912 | |
| | Random Forest | 0.852 | | 0.820 | | 0.870 | | 0.911 | |
| | AdaBoost | 0.867 | 0.866 | 0.830 | 0.834 | 0.890 | 0.886 | 0.898 | 0.910 |
| | Gradient Boosting | 0.872 | | 0.840 | | 0.890 | | 0.916 | |
| | Extra Trees | 0.872 | | 0.840 | | 0.890 | | 0.914 | |

The results in Table 2 show that the best scaling methods based on the score obtained in machine learning metrics are standardized and MaxAbs. Therefore, the focus of ensuing analyses will be placed on these two.


Noticeably, the best performing classifiers all belong in the family of tree-based algorithms. Decision Tree (DTC), Random

Forest (RFC), AdaBoost Classifier (ABC), Gradient Boosting (GBC) and ExtraTrees (ETC) consistently outperformed other

classification methods throughout the process. While tree-based algorithms do not necessarily require feature scaling (Li et

al.,2017), the outcomes of this experience show that different scaling methods lead to slightly different performances. The best

test scores were obtained by the ETC algorithm (0.878 and 0.872, for standardize and MaxAbs methods respectively), while

the test scores of the other algorithms remained above 0.85. The F1-score for the positive groundwater potential is above 0.87

in all cases, while the F1-score for the negative potential decreases slightly (0.82 to 0.85). AUC scores oscillate between 0.89 and 0.91, which reinforces the idea that the best-performing classifiers are able to accurately predict the target.

By their own nature, tree-based algorithms can be expected to yield good results in groundwater potential studies. DTC is a popular technique, largely due to the ease with which the internal logic of the algorithm can be interpreted (Naghibi et al., 2016; Moghaddam et al., 2020). In turn, RFC is a combination of such tree predictors that each tree depends on the values of a random vector sampled independently and with the same distribution for all trees in the forest (Breiman, 2001). RFC is one of the most powerful machine learning classifiers in general, and typically renders useful outcomes in predictive mapping

(Naghibi et al., 2017b; Martínez-Santos and Renard, 2020). The GBC algorithm is an ensemble method in which multiple weak classifier trees are used to build a strong classifier (Friedman, 2001). In the same way, the basic principle of ABC (Freund and Schapire, 1997) is to fit a sequence of weak learners into repeatedly modified versions of the data. The predictions of all of them are combined by a weighted majority vote to produce the final prediction. The data modifications at each iteration of the so-called boosting consist of applying weights to each of the training samples (Pedregosa et al., 2011). The literature shows

that GBC and ABC have been successfully used as ensemble methods in the development of GPM maps (Nguyen et al., 2020a; Martínez-Santos and Renard, 2020). Finally, for the ETC algorithm (Geurts et al., 2006), randomization goes a step further in the way splits are computed. As in random forests, a random subset of candidate features is used, but instead of looking for the most discriminative thresholds, random thresholds are extracted for each candidate feature and the best of these randomly generated thresholds is chosen as the splitting rule. This usually allows the variance of the model to be reduced somewhat

further (Pedregosa et al., 2011).

Other algorithm families have been successfully applied in GPM research. Support vector machines (Martínez-Santos and Renard, 2020), logistic regression (Ozdemir, 2011; Chen et al., 2018), neural networks (Moghaddam et al., 2020; Panahi et al., 2020) have all been demonstrated to be efficient predictors of groundwater potential, even when pitched against tree-based

algorithms. This suggests that differences in input databases and the choice of explanatory variables may constrain the applicability of each algorithm in each given context. Furthermore, the inherent complexity of supervised classification makes it difficult for users to understand the internal logic of the algorithms, which means it is frequently impossible to predict which one will perform best in a given situation. Incorporating various algorithm families in GPM is therefore perceived as a suitable course of action in GPM studies.


**Table 3.** Algorithm hyperparameters (description after Pedregosa et al., 2011) and adjusted outcomes. Max_depth = The maximum depth of the tree; max_features = The number of features to consider when looking for the best split.; min_samples_leaf = The minimum number of samples required to be at a leaf node; min_samples_split = The minimum number of samples required to split an internal node; random_state = Controls both the randomness of the bootstrapping of the samples used when building trees and the sampling of the features

to be considered when looking for the best split at each node.; n_estimators = Number of trees in the ensemble methods; algorithm = method

used by Ada Boost (discrete SAMME boosting algorithm or real SAMME.R boosting algorithm); learning_rate = shrinks the contribution of each classifier.

| Classifier | Hyperparameter | Optimized parameter value | |
| --- | --- | --- | --- |
| | | Standardize method | MaxAbs method |
| Decision Tree | max_depth | 3 | 6 |
| | max_features | 0.9 | 0.6 |
| | min_samples_leaf | 8 | 5 |
| | min_samples_split | 0.1 | 0.1 |
| | random_state | 0 | 0 |
| Random Forest | max_depth | 6 | 5 |
| | max_features | 1.0 | 1.0 |
| | min_samples_leaf | 8 | 8 |
| | n_estimators | 141 | 196 |
| | random_state | 0 | 0 |
| Ada Boost | Algorithm | 'SAMME' | SAMME.R |
| | learning_rate | 0.795 | 0.109 |
| | n_estimators | 340 | 170 |
| | random_state | 0 | 0 |
| Gradient Boosting | max_depth | 2 | 2 |
| | max_features | 0.6 | 0.2 |
| | min_samples_leaf | 20 | 20 |
| | n_estimators | 100 | 160 |
| | random_state | 0 | 0 |
| Extra Trees | max_depth | 9 | 8 |
| | max_features | 0.8 | 0.8 |
| | min_samples_leaf | 1 | 5 |
| | n_estimators | 520 | 350 |
| | random_state | 0 | 0 |

## 3.3 Importance of explanatory variables

Naghibi and Pourghasemi (2015) indicate that the importance of explanatory variables in groundwater potential mapping is
considerably influenced by the approach used in an investigation and the study area properties. Outcomes show that elevation, rainfall, geology and drainage density, among others, are the most important factors conditioning the groundwater potential,

which is in agreement with results obtained by other authors in different geographical contexts (Ozdemir, 2011; Naghibi and Pourghasemi, 2015; Nguyen et al., 2020b).

A major advantage of incorporating recursive feature elimination is that it eliminates part of the expert bias associated with the choice of explanatory variables. In this case, the fact that all variables are used by at least two of the best-performing algorithms suggests that the initial choice of explanatory variables was appropriate. However, feature selection reveals clear differences among the classifiers. Under standardized scaling, RFC only required three explanatory variables to predict groundwater potential (precipitation, expected saturated thickness and elevation). ABC and GBC used eight each, while ETC

and DTC used eleven and fourteen, respectively. Under MaxAbs scaling, the RFC and ABC used three and four variables, respectively. GBC algorithm worked with six, DTC used sixteen and ETC seventeen. The variables used by each of the best-performing algorithms are presented in Figure 9.

All algorithms agree on the importance of elevation and precipitation. The combined importance of these two variables for the

RFC, GBC, ABC, DTC and ETC algorithms is 0.90, 0.71, 0.47, 0.90, and 0.42, respectively, under the standardize scaling method. For the MaxAbs method, the combined weights amount to 0.90, 0.51, 0.64, 0.87 and 0.47, respectively. Other commonly used features include the expected saturated thickness (eight out of ten cases), drainage density (six), NDWI (six), geology (five), clay content (five) and NDVI (five). The relative weight of other variables varies depending on the algorithm and scaling method, but most agree on the importance of expected saturated thickness, slope, geology and drainage density.

The least used variables are the alteration layer, curvature, soil and SPI.

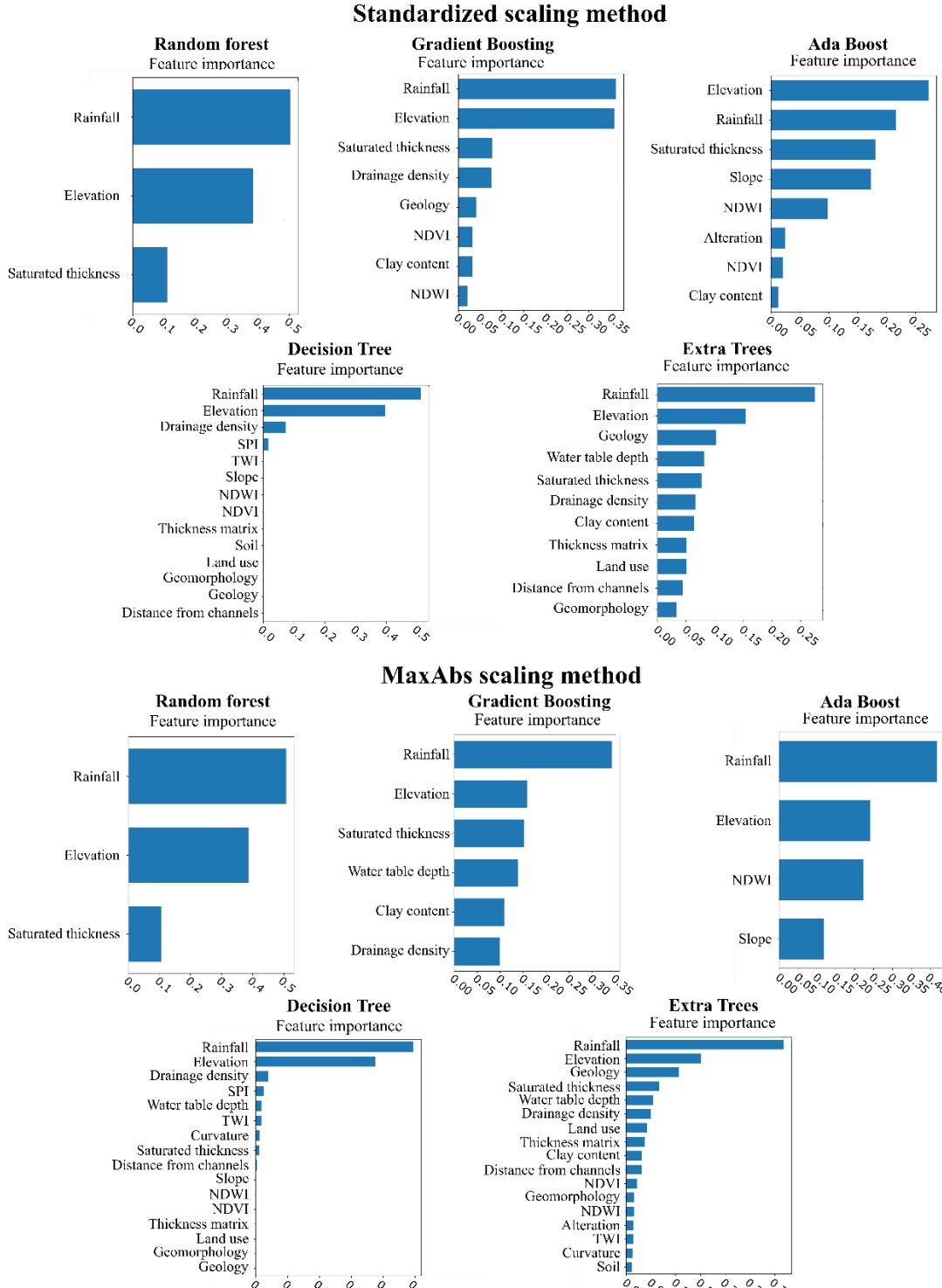

**Figure 9.** Feature importance calculated for the best performing tree-based algorithms using the standardized and MaxAbs scaling methods. The sum of all variable weights equals one.

### 3.4 Groundwater potential maps

Classifier outcomes were extrapolated to produce groundwater potential maps. Figure 10 shows the groundwater potential predictions rendered by each of the five best-performing algorithms under the two most effective scaling methods. Red areas are those in which the algorithms have found a combination of explanatory variables leading to a negative potential. In turn, green zones represent a positive groundwater potential. All maps show a gradient characterized by the predominance of positive areas in the south to a greater proportion of negative areas in the north. This appears to be related to rainfall patterns.

Other major negative zones occur around mountain outcrops of the southwest. Low potential areas occur around mountain outcrops of the southwest. The large green zone around the southern region corresponds to the weathering mantle of basement rocks. Groundwater in basement aquifers is most often found in weathered formations and piedmonts of the outcrops. Piedmonts may exhibit high GPM because these are essentially a mixture of weathered and transported materials (Martín-Loeches et al., 2018). This area presents smooth orography, which means that high GPM is mainly determined by the weathered

mantle. Previous research by Diaz-Alcaide et al. (2017) attributes a medium GPM for the southern part of the Koulikoro region. Discrepancies between this and our results likely stem from the fact that these authors used a commune-based approach, rather than physical variables.

The central part of the agreement map shows a high potential for both methods, except in the higher altitude areas. This region,

consisting of consolidated metasedimentary materials, has an average aquifer thickness of 30 to 50 metres (Traore et al., 2018). High yields are associated with the weathered mantle developed in the upper part instead of with the predominant lithology. This becomes evident in the metasedimentary outcrops located in the highlands. These present a low groundwater potential because fracturing facilitates rapid groundwater percolation into the plains, where it accumulates in the alteration zone. Areas near major rivers, such as the Niger, Sankarani and Bani, also have a high potential. This is attributed to the high permeability

of alluvial sediments. The northern part of the study area, formed by consolidated and unconsolidated sedimentary materials, has a low groundwater potential. Although geological conditions are more favourable for groundwater due to the type of materials than in other areas of the region, groundwater potential is limited by low rainfall. This is demonstrated by the feature importance analysis, which shows that precipitation is one of the two most important explanatory variables for groundwater occurrence.


Overall, maps are remarkably similar, although the MaxAbs scaling method renders a slightly greater proportion of negative potential areas. The agreement map (Fig. 11) allows for an analysis of discrepancies among the best performing algorithms. The MaxAbs scaling method predicts a low groundwater potential (pixel value 0-0.2) for 41% of the study area, while about 40% is identified as a high groundwater potential (>0.8) and the remaining 19% consists of zones with moderate potential (0.2

- 0.8). The standardized scaling method renders a high groundwater potential for 42%, low groundwater potential for 38%, and various degrees of moderate potential for 20% of the region.

# Individual maps

**Groundwater potential**

🟥 0 - Negative
🟩 1 - Positive

## MaxAbs scaling method

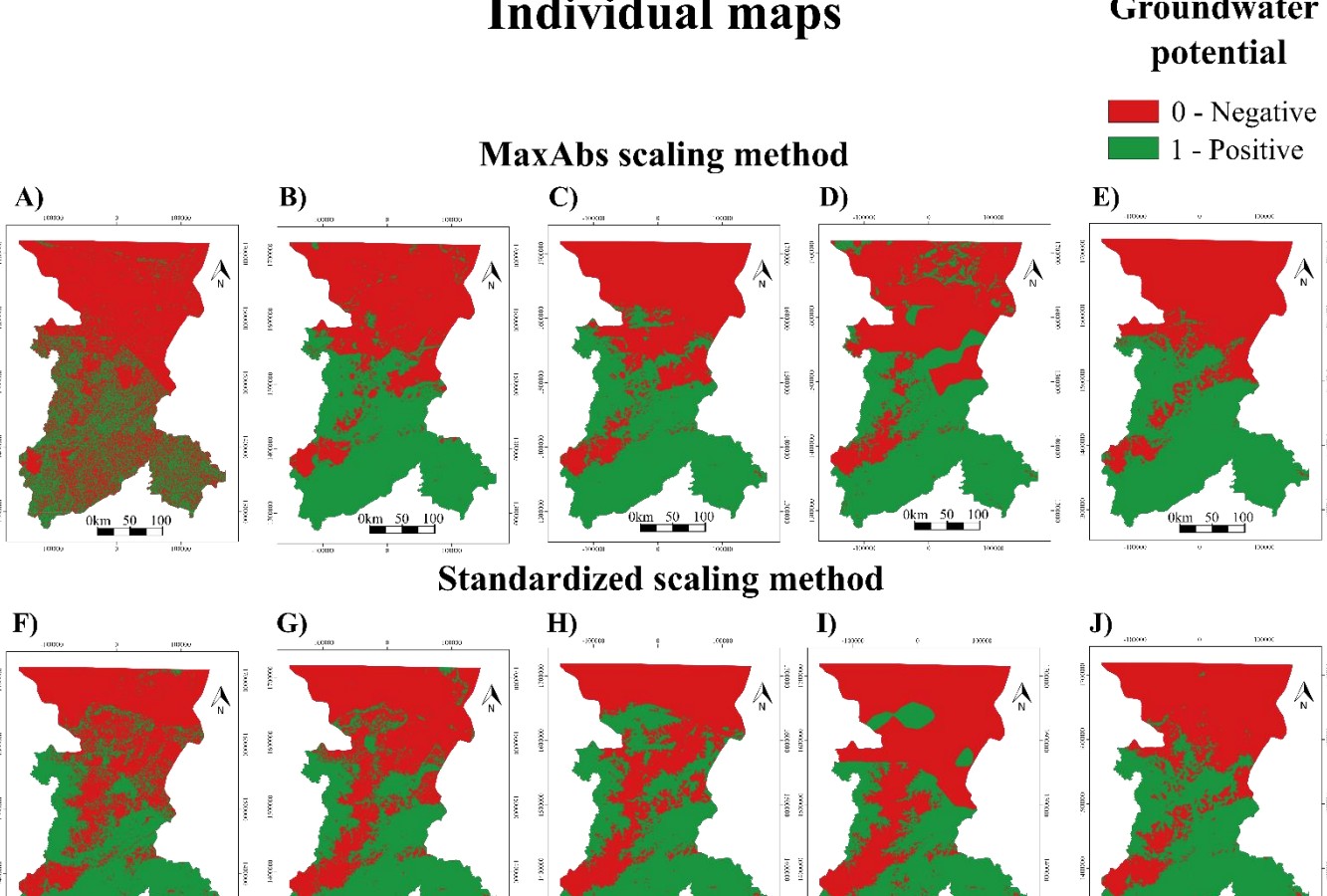

## Standardized scaling method

**Figure 10.** Mapping outcomes of the top five supervised classification algorithms for the two best performing scaling methods. At the top the MaxAbs scaling method: A) AdaBoost classifier B) Gradient Boosting classifier C) Random Forest classifier D) Decision Tree classifier E) Extra Trees classifier. Below it, the standardized scaling method: F) AdaBoost classifier; G) Gradient Boosting classifier H) Random Forest classifier I) Decision Tree classifier J) Extra Trees classifier.

Martínez-Santos et al. (2021a) show that standard machine learning metrics do not necessarily provide an unambiguous measure of classifier performance when predicting spatially-distributed outcomes. These authors argue that case-specific indicators could provide additional insights on the results. In this case, an independent dataset of groundwater data (borehole yield) was used to cross-check machine-learning predictions. Villages featuring more than five boreholes (n=334) were relied upon for this purpose. These villages were classified into three groups based on the predicted groundwater potential ("low" <0.2; "moderate" 0.2 to 0.8; and "high" >0.8). Table 3 shows the mean borehole success rate, the mean borehole yield, and the percentage of boreholes with a flow rate in excess of 10 m$^3$/h in each category. The average water table depth is also provided for reference.

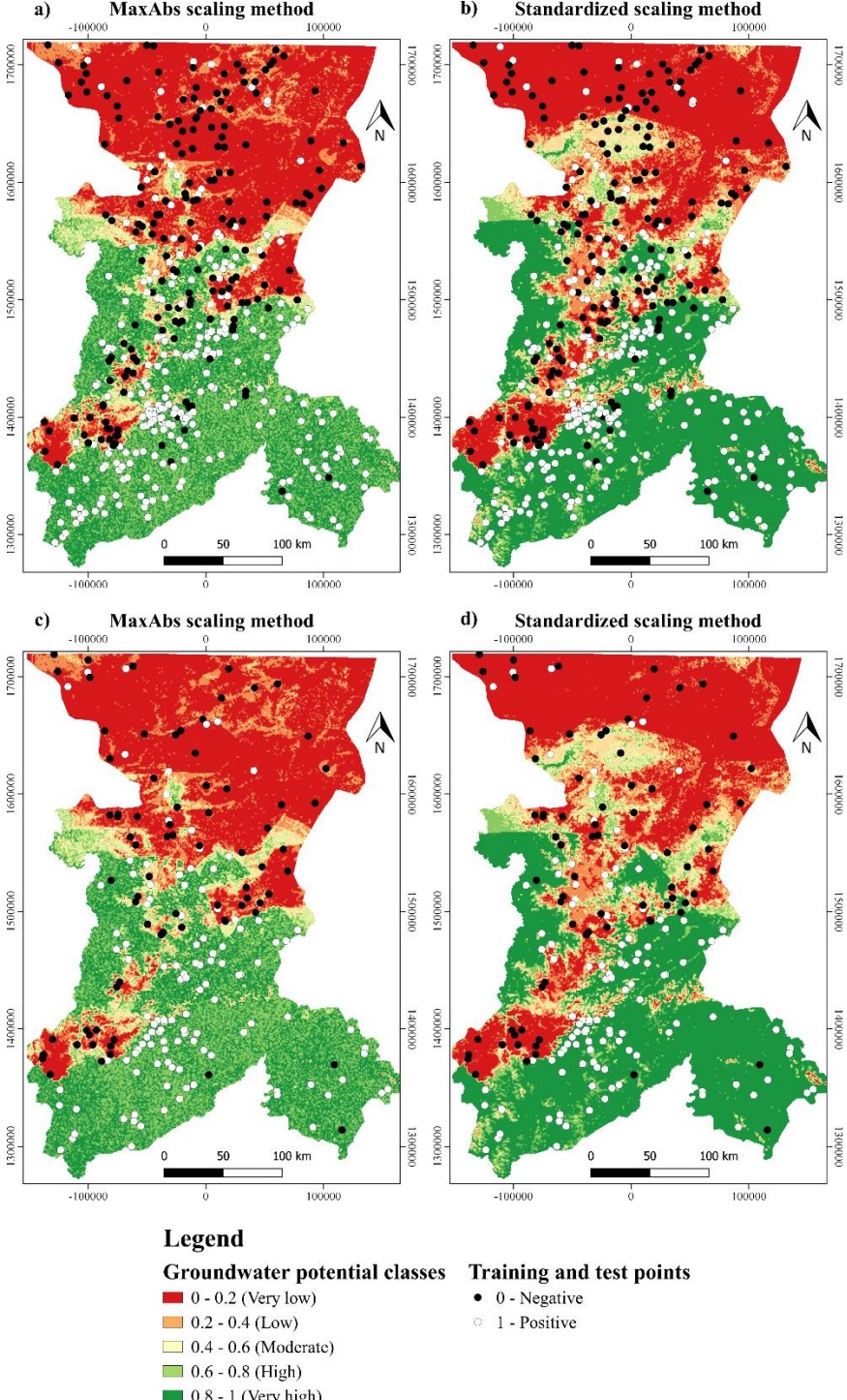

**Figure 11.** Mapping outcomes of the agreement map for MaxAbs scaling method and Standardized scaling method. a) Training points on the MaxAbs scaling method agreement map. b) Training points on the Standardized scaling method agreement map. c) Testing points on the MaxAbs scaling method agreement map. d) Testing points on the Standardized scaling method agreement map.


A first conclusion that springs to mind is that about 60% of the villages with more than five boreholes fall within the high groundwater potential description, while a further 20% present a moderate groundwater potential. To a certain extent, this was to be expected, as the larger populations develop in areas where groundwater is available. Table 3 also shows groundwater potential outcomes to match borehole flow rates reasonably.

The mean borehole success rate in areas labelled as high groundwater potential is 75.9% for the MaxAbs scaling method and 73.7% for the Standardized method. In contrast, the mean success rate for villages in the low groundwater potential category is 33.6% and 32.1%, respectively. These outcomes suggest that both scaling methods and, in general, the machine learning approach are adept at identifying areas where boreholes are more likely to be successful.

Both the mean borehole yield and the percentage of high-yield boreholes ($>10 m^3$/h) provide interesting benchmarks for comparison. The mean borehole yield is consistent with map outcomes. Under the MaxAbs scaling method, the mean borehole yield is 1.39 $m^3$/h in the villages located within low groundwater potential areas and 3.92 $m^3$/h in areas where groundwater potential is predicted to be high. Furthermore, 76.1% of the high-yield boreholes fall within high groundwater potential areas, while just 6.6% are located in low groundwater potential areas. Results are similar for the standardized scaling method. The
mean borehole yield is 1.44 $m^3$/h and 3.63 $m^3$/h for the low and high potential categories, respectively, while 74.6% of the high-yield boreholes are located in villages within high groundwater potential zones.

**Table 4.** Results of the analysis based on the agreement map categories for the MaxAbs and Standardized scaling methods. The analysis was performed for villages with more than 5 wells. The average success rate, average flow rate, average flow rate with the negative boreholes
and water table depth as well as the percentage of wells with a yield higher than 10$m^3$/h are shown.

| Groundwater potential | Number of villages | Borehole success rate (%) | Mean borehole yield (m³/h) | Mean borehole yield - only positive boreholes (m³/h) | % High yield boreholes (>10 m³/h) | Average water table depth (m) |
|---|---|---|---|---|---|---|
| MaxAbs scaling method | | | | | | |
| Low | 70 | 33.6 | 1.39 | 3.58 | 6.6% | 18.6 |
| Moderate | 69 | 50.5 | 2.23 | 4.36 | 17.3% | 15.1 |
| High | 195 | 75.9 | 3.92 | 5.10 | 76.1% | 12.1 |
| Standardized scaling method | | | | | | |
| Low | 65 | 32.1 | 1.44 | 3.96 | 6.6% | 19.3 |
| Moderate | 70 | 55.5 | 2.78 | 4.57 | 18.8% | 16.1 |
| High | 199 | 73.7 | 3.63 | 4.87 | 74.6% | 11.8 |

On a final note, the literature features few examples of groundwater potential studies in the study area. Perhaps the only systematic precedent is that carried out by Díaz-Alcaide et al. (2017). These authors performed a national-scale assessment of groundwater potential for the Republic of Mali based on the same borehole database that has been used in this research. However, they aggregated their data at the commune scale rather than at the village scale, and relied on different classification principles. Their outcomes are thus to be interpreted at a different spatial resolution. This notwithstanding, their groundwater potential assessment for the Bamako and Koulikoro regions is similar to that obtained by means of machine learning classification. The lower groundwater potential areas were identified in the north and the higher groundwater potential ones were found mostly in the south. This attests to the capacity of machine learning approaches to realistically depict field conditions, and thus, to underpin groundwater development.

### 3.5 Limitations

Predictive groundwater mapping presents some uncertainties. These are typically associated with deficiencies in data quality (e.g., small sample sizes, missing covariates, biased and missing data), as well as with errors in the structural nature and specifications of the model (Rahmati et al. 2015). In spatial modelling studies the sample size has proven to be a significant factor affecting the predictive abilities of the models (Guisan et al., 2007). Moghaddam et al. (2020) analysed the influence of sample size on GPM, concluding that there is a significant decrease in AUC values when the sample size is 25% of the input dataset. In the present case, with 650 ground-truth points, the effect of sample size could be ruled out as per this standard.

Input data was made available by government officers after careful field evaluation. From this perspective, the database is considered to be of high quality. However, it is also true that key hydrogeological variables are missing. Groundwater potential was evaluated in binary form (positive/negative) because the borehole database provides little information in terms of borehole productivity. Spatially distributed estimates of aquifer transmissivity, storage coefficients and yield would allow for a multi-class assessment, which in turn would provide a more realistic zoning of groundwater potential. Besides, the village scale resolution presents some shortcomings. As per the database, all boreholes within a given village have the same coordinates. This makes it difficult to train the algorithms when, for instance, the village overlies a non-homogeneous hydrogeological setting. A more detailed knowledge of how the boreholes are distributed in space would be expected to enhance the outcomes.

Along the same lines, the resolution of certain explanatory variables is potentially problematic. Take for instance soil and lithology, which are only available at the regional scale, and which might constrain groundwater potential to an important extent. Gómez-Escalonilla et al. (under review) explain that inroads can be made in the use of a dynamic explanatory variable. Take for instance evapotranspiration or seasonal fluctuations VV- and VH-polarization intensity (backscattering coefficient) and VV- polarization coherence (interferometric correlation) from the Sentinel-1 time-series from which temporal descriptors are derived. Furthermore, Worthington (2015) shows that groundwater modelling in bedrock aquifers is complex because there is often a substantial flow through fractures, and the interconnectivity and magnitude of these fractures are usually uncertain.

In this context, geophysical techniques can provide useful information that improves the ability to predict GPM. However, the absence of geophysical information also limits our knowledge about subsurface structures.

The lack of interpretability is a major drawback of machine learning algorithms. Except for the simple decision tree, whose internal logic is straightforward, all classifiers work according to a complex internal architecture. Therefore, it is nearly

impossible to understand the reasoning that leads most algorithms to a given conclusion beyond computing feature importance. This in turn results in extensive trial and error before optimal outcomes are achieved.

Overfitting can lead to spurious results in machine learning models. Overfitting occurs when the model does not generalize well with the training data (i.e. it tries to fit the training data perfectly at the risk of missing the underlying associations between

the explanatory and target variables). To address this, techniques such as splitting the initial data set into separate training and test subsets, cross-validation, regularization and ensembling may be used (Dietterich, 1995; Yeom et al., 2018). Case-specific indicators may also be used to appraise the results beyond standard machine learning metrics, thus adding to the robustness of predictions (Martinez-Santos et al 2021a and 2021b). In the case at hand, this is achieved by comparing map outcomes with the limited available data on borehole yield.

**4 Conclusions**

Machine learning applications are gaining recognition as valuable tools to underpin groundwater management. The ease with which machine learning algorithms find complex associations within large datasets, together with their ability to develop accurate predictions, open up a whole new dimension to the analysis of groundwater data. Artificial intelligence approaches may improve water access in areas such as the Sahel by providing additional guidance to borehole drilling initiatives. Within

this context, our research combined state-of-the-art machine learning tools, including collinearity checks, recursive feature selection, feature importance computation and cross-validation mechanisms to map groundwater potential in two regions of Mali. Tree-based algorithms consistently outperformed other algorithm families such as support vector machines and neural networks. Since this cannot be expected to occur in every case, we advocate the use of a large number of machine learning classifiers and the subsequent selection of the best performers as a sensible course of action. The same logic entails that a range

of scaling methods should be used to prevent expert-based bias associated with the reclassification of explanatory variables.

A crucial finding of this research is that conventional machine learning metrics (test score, area under the receiver operating characteristic curve) can be more representative of algorithm performance than of the actual field conditions. This is particularly relevant when attempting to develop spatially-distributed predictions. Double-checking algorithm results with an

independent groundwater dataset (borehole flow rates in this case) is recommended to ensure that map outcomes are accurate. Machine learning approaches are thus seen as a means to underpin borehole siting initiatives at a the regional scale, although

it is recognized that local-scale fieldwork is needed for optimal outcomes. In the context of Sustainable Development Goal #6, predictive groundwater maps such as those developed in the course of this research may be of use to private investors willing to participate in improving water access in remote regions, as well as to government officers, cooperation funds and international donors.

From a regional perspective, groundwater potential closely resembles rainfall. Despite the predominance of low-permeability basement rocks, medium to high groundwater potential observed in southern areas seems to be associated with high rainfall and well-developed weathering mantles. In contrast, theoretically good conditions for groundwater storage in the north present a low potential due to limited precipitation. The central part is characterized by a medium groundwater potential in the plains, as well as by a high potential in alluvial sediments of the major river systems and low potential in the highlands.

*Author contribution*

VGE: Conceptualization, Formal analysis, Methodology, Software, Writing – original draft preparation. PMS: Conceptualization, Methodology, Software, Funding acquisition, Writing – review & editing. MML: Conceptualization, Writing – review & editing

*Competing interests*

The authors declare that they have no conflict of interest.

*Acknowledgements*

This work has been funded under research grant RTI2018-099394-B-I00 of Spain's Ministry of Science, Innovation and Universities. The first author received an FPI grant from the Ministry of Science and Innovation to develop his PhD within this project (PRE2019-090026). The second author received a Salvador de Madariaga grant (PRX18/00235) from Spain's Ministry of Education, Culture and Sport to carry out a 3-month research stay at the Université de Neuchâtel, Switzerland, where the original version of the software used in this paper was developed. The authors thank the Direction Generale de l'Hydraulique of Mali for making its borehole database available.

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
