# Peer review of "Preprocessing approaches in machine learning-based groundwater potential mapping: an application to the Koulikoro and Bamako regions, Mali"

_Hydrology and Earth System Sciences, 2021_

## Referee Comment (RC1)

**Referee comment to:**

HESS-2021-261 Revision report

"Preprocessing approaches in machine learning-based groundwater potential mapping: an application to the Koulikoro and Bamako regions, Mali" by Gómez-Escalonilla et al. (2021).

**General Comments**

Gómez-Escalonilla et al. (2021) provide an interesting machine learning learning-based groundwater potential mapping in the Koulikoro and Bamako regions of Mali.

The authors have used machine learning models for groundwater potential mapping (GPM) in two regions in Mali and evaluated their models. Also, their models explore the potential factors affecting groundwater potential.

The paper is interesting and within the scope of the HESS journal. In general, machine learning is well-placed in HESS. The authors have done very diligent work by summarizing many publications applying machine learning. The manuscript can be interesting to the scientific community working on machine learning applied in hydrology. The work is of importance; but at the present state, I would not recommend it for publication because certain comments need to be addressed with major revisions.

- The introduction is very general. It should be worked out why this study with machine learning is necessary, knowing that machine learning is a "Blackbox model" and what its benefit is with other methods such as fuzzy logic, the frequency ratio, weight of evidence, or multi-criteria decision analysis (MCDA). In addition, the objectives are not clear and are included at various locations in the manuscript (see Page 13, Line 255- 260, with the sentence "A major goal of this study…"). You need to improve the manuscript correctly.

- In the Introduction section only studies for the other continent are presented. It would be interesting to see how studies in other regions in Africa deal the groundwater potential mapping with machine learning or others methods? Also, the authors should be linking the issues of groundwater resources in the context of The Sustainable Development Goals (SDGs) to motive the reader at the end of the introduction.

- You could add this reference in the introduction section: A new method to map groundwater potential at a village scale, based on a comprehensive borehole database. An application to Sikasso, Republic of Mali by Ana Carolina Gonçalves Delgado, 2018.

- Add a new section (or put some sentences) about the limitations of machine learning techniques to study groundwater potential zones.

- The overfitting problem is one of the drawbacks that affect the accuracy of models in machine learning. Take into account this issue in the introduction.

- The topic of validation of Groundwater Potential Map by is not mentioned. In my opinion, it is one of the major limitations of the study. If so, this topic should be discussed in more detail.

- Did you limit the validation of your model with cross-validation? Or do you have the intention to integrate the external validation?

- The authors need to describe in the methodology section a sub-section of "Multicollinearity analysis" before presenting the results in section 3.1.

- Did you use the variance inflation factor (VIF) and tolerance (TOL) indices as are customarily used to estimate multicollinearity of predictive factors in machine learning modelling? If so, explain in the manuscript.

- What is the effect of sample size on the different machine learning models for groundwater potential mapping in your research?

- Did you try to make a sensitivity analysis of the effect of each factor (explanatory variables) on the groundwater potential map, i.e., when you decide to eliminate one or more factors??

- What is the resolution chosen to develop the thematic layers? Because the various GIS layers come with different spatial resolutions. You need to clarify this aspect.

- The sub-sections in Section 2 on "Material and methods" need to be reorganised for better reading. For example, you could define the title "Materials and methods": Study area; Data used (Borehole database, Explanatory variables/Thematic layers, etc); and Methods (Random Forest, AdaBoost, Gradient Boosting, Decision Tree, and Extra Trees classifiers); Tools used to process the data, etc.

- A table of data sources must be put to increase the clarity and to ease readers' understanding.

- Do you have performed quality control of datasets before the modelling?

- Did you use all explanatory variables to map groundwater potential in Figure 10? If so, could you specify the variables used to develop the final products? Because, on Page 20 of MS, you mentioned that the outcomes show that elevation, rainfall, geology, and drainage density, among others, are the most important factors conditioning the groundwater potential. Did you use these four (04) explanatory variables in reality? Specify exactly the variables used in the final models.

- The conclusion is very general. To be check according to the revision of MS.

- Limitations of the research should be addressed.

- I suggested the authors separate the results and discussion.

- I suggest developing in the new discussion section "the model validation/performance and comparison"; "assessment of variable importance"; limitations of the research", etc. Furthermore, try to compare the outcomes of your research with other studies available in the literature on the mapping of groundwater potential such as the GIS-based Dempster–Shafer model, etc.

- The discussion is incomplete, authors must address the uncertainty in groundwater potential mapping (deficiencies of data quality; biased and absent data, sample sizes, missing covariates, and also errors in the structural and nature of the model, etc). Add some references.

- In the Conclusion section, specify the utility of this research and the potential users. For example, who could use the prepared maps?

- Is it possible to improve the performance of the best machine learning models in your study? Which additional predicting variable (s) (even if such information is scarce) could be added to improve the results?

- I think that Table 2 on Page 19 could be placed in a new section in Supporting Information.

- **Abstract section:**

- Page 1. The abstract is very long. It should be shortened and focused.
**Keywords**: I propose to delete "big data; climate change and water access"; and add "Groundwater potentiality, and GIS".
- The abstract should be thoroughly revised according to the revision of the manuscript.

**Specific comments:**

Page 2. Paragraph 1. Line 2, rewrite the sentence "Today, 2.5 billion people…." by "Today, 2.5 billion people around the World…".

Page 2. Paragraph 3. Line 1. Introduce a sentence before this: "There are two main approaches to GPM: expert-based decision systems and machine learning methods".

Page 3. For section 2.4, Material and methods, I suggest separating them into two sections.
2.4.1 Definition of target
2.4.2.2 Explanatory variables/Thematic layers
In this new section, I propose to describe the explanatory variables by order according to Figure 6.
Also, I propose to prepare explanatory variables used in groundwater potential mapping in a Table. In your Table, you could, for example, put in 4 columns (Type of data layers/ Explanatory variables; scale/resolution, time, available format, and source of data).

Page 6. Line 3. Specify correctly if the numbers 530 and 452 are the number of villages?

Page 9: Put in order Figure 5a; before Figure 6.
Page 9: Rewrite the last sentence by also, it was used to.

Page 9. Figure 4 must be centered.

Page 12: Fig.12: order the number of figures following the description found on page 9, i.e.: curvature, slope, topographic wetness index (TWI),

Page 13: Number 2.5 was repeated on page 14. Check it. I have the impression that the authors did not take the time to proofread the document.

Page 13: 2nd paragraph. The objectives of the study mentioned at various locations in the manuscript should be summarized at the end of the introduction (see comments above). Why did you put the main goal of the study here? I think that the objective must be found in the introduction section.

Page 14. 3rd paragraph. Did you fix the number of iterations at 500 in this study? It is the default value of model? Justify how this number was established.

Page 17: you mentioned in the first line that "The AUC exceeds 0.90 in all cases". I'm not sure about this affirmation because, if you analyse Table 1, you observe that in the MaxAbs scaling method AdaBoost shows an AUC value of 0.898. Could you rewrite your sentence to take into account this case? Or maybe use AUC mean because this value exceeds in all cases.

Page 20. First-line (Line 415). You mentioned Naghibi and Pourghasemi (2015), the citation is incorrect because you have three authors: Seyed Amir Naghibi & Hamid Reza Pourghasemi & Barnali Dixon.
At the end of this paragraph again, you mentioned (Naghibi and Pourghasemi, 2015; Nguyen et al. 2020b). Due to this error for the citation in two places, I propose to check all references.

Page 21: Move Figure 8 on page 21 under the section of "3.3 Importance of explanatory variables" on
Page 23: I propose to add the well locations/boreholes on the two maps in Figure 8.
If possible, put on these two maps: well training and well validation with different colours of points.
Also, make clear your legend of Fig.8 with the classes well-defined.
For example:
 (0- 0.2) Very low;
 (0.2- 45) Low;
Etc.
Change the term "Intermediate" in the legend by "Moderate". It is most appropriate.
Change "Groundwater potential" to "Groundwater potential classes".

Page 21. I repeat the need to clarify my request mentioned above (see general comments). When you analyse feature importance calculated in Figure 8, you observe that some explanatory variables are not important in the models. Could you explain more how many variables did you select to produce the outcomes of Figure 10?

Page 23. Why did you choose to classify villages in three classes based on groundwater potential, and you show the outcomes of Groundwater potential in five classes?
Page 24. Could explain more why Groundwater potential classes are three in Table 3 compared to Figure 10, where we found five classes?

Page 15. Section 3 on **Results and discussion**. Please add a new sub-section on "Validation on machine learning models.

**Reference section**

Page 27.   Rewrite this reference: Direction Nationale de l'Hydraulique (Ed.): Données Hydrogeologiques et des Forages. Direction Nationale de l'Hydraulique du Mali, 2010.
 to the precise country name.
Page 29.  Precise the link and access date of this reference: Poggio, L. and de Sousa, L.: SoilGrids250m 2.0 - Clay content, 2020.

Page 30.
- Add the access date of the reference **Traore, A, Z., et al.**
- Add the link and access date of this reference: United Nations: Resolution A/RES/64/292. United Nations General Assembly, United Nations, 2010.

**Technical corrections**

Page 5. Line 115-120. Add the unity of mean water depth to be coherent in the sentence, because you have put the unity of mean electric conductivity.
Page 6: in Figure 2 B, write correctly $m^3/h$
Page 8. Line 160-165, add the comma in (BGS, 2021). Also, on Page 9 and the title of Figure 4, add a comma to the same reference.
Page 11, line 5. "semiarid" "semi-arid"

Page 13:  Equation 6; define   $\tilde{x}$
Page 25. Delete "s" in Conclusion.

---

## Author Comment (AC1)

**Referee comment to:**

**HESS-2021-261 Revision report**
**"Preprocessing approaches in machine learning-based groundwater potential mapping: an application to the Koulikoro and Bamako regions, Mali" by Gómez-Escalonilla et al. (2021).**

**[1] General Comments**
**Gómez-Escalonilla et al. (2021) provide an interesting machine learning learning-based groundwater potential mapping in the Koulikoro and Bamako regions of Mali. The authors have used machine learning models for groundwater potential mapping (GPM) in two regions in Mali and evaluated their models. Also, their models explore the potential factors affecting groundwater potential. The paper is interesting and within the scope of the HESS journal. In general, machine learning is well-placed in HESS. The authors have done very diligent work by summarizing many publications applyingmachine learning. The manuscript can be interesting to the scientific community working on machinelearning applied in hydrology. The work is of importance; but at the present state, I would not recommend it for publication because certain comments need to be addressed with major revisions.**

We thank Reviewer #1 for a series of insightful comments. We have strived to incorporate them all to the manuscript. We believe this has helped us improve the quality of our work.

**[2] The introduction is very general. It should be worked out why this study with machine learning is necessary, knowing that machine learning is a "Blackbox model" and what its benefit is with other methods such as fuzzy logic, the frequency ratio, weight of evidence, or multi-criteria decision analysis (MCDA).**

Agreed. A key point is that methods such as Weight of evidence, Frequency Ratio and Multi-criteria decision analysis require a grouping of the variables in intervals. A bias is generated from the outset, since these intervals rely almost exclusively on expert criteria. We attempt to show that, if pre-processing is involved, machine learning algorithms can work directly with raw data, thus discarding a potentially biased clustering of explanatory variables. Another important advantage of supervised classification is that the power of artificial intelligence can be harnessed to find complex associations among explanatory variables that might otherwise pass unnoticed

In order to incorporate this comment to the manuscript, the third paragraph of the introduction has been rewritten as:

*"The literature shows that there are two main approaches to GPM, namely, expert-based decision systems and machine learning methods. Expert-based system methods have been used for a long time (DEP, 1993). These include multi-influence factor techniques (Magesh et al., 2012; Nasir et al., 2018; Martín-Loeches et al 2018), analytical hierarchy processes (Mohammadi-Behzad et al., 2019; Al-Djazouli et al., 2021), and Dempster-Shafer models (Mogaji and Lim 2018, Obeidavi et al 2021). Other frequently used methods are weight of evidence and frequency ratio analysis (Falah and Zeinivand, 2019; Boughariou et al., 2021). Machine learning is comparatively newer. A major difference between machine learning and expert approaches is that supervised classification uses the advantages of artificial intelligence to find complex associations among explanatory variables that might otherwise pass unnoticed. Hence, machine learning is well suited to map complex spatially-distributed variables such as groundwater occurrence. The GPM literature showcases a wide variety of supervised classification approaches. Thus, Al-Fugara et al. (2020) used mixed discriminant analysis to map spring potential in a watershed of Jordan; much like Odzemir (2011) mapped spring potential in a Turkish basin by means of a logistic regression method. Random forests have proved adept at mapping groundwater potential, both in mountain bedrock aquifer (Moghaddam et al., 2020), as well as in large metasedimentary basins (Martínez-Santos and Renard 2019). Other supervised classification methods include boosted regression trees (Naghibi et al., 2016), support vector machines (Naghibi et al., 2017b), neural networks (Lee et al., 2012; Panahi et al., 2020) and Ensemble methods (Naghibi et al., 2017a; Martínez-Santos and Renard, 2019; Nguyen et al., 2020b)."*

**[3] In addition, the objectives are not clear and are included at various locations in the manuscript (see Page 13, Line 255- 260, with the sentence "A major goal of this study…"). You need to improve the manuscript correctly.**

Agreed. In order to clarify the goals we have rewritten some sentences of the last paragraph of the introduction, which now reads:

*"The outcomes of machine learning GPM studies are almost invariably assessed by means of standard big data metrics such as precision, recall, and area under the receiver operating characteristic curve. While useful, these are of limited value in cases where the input dataset consists solely of unambiguous examples. Furthermore, there are question marks as to whether these metrics are truly representative for the development of spatially-distributed estimates (Martínez-Santos et al., 2021). In those instances, using ad hoc calibration elements, such as complementary field information, can contribute to a better interpretation of the outcomes. The objective of this research is to build on the existing literature by presenting two main methodological additions. In the first place, we explore different scaling methods in order to avoid the pitfalls associated with the reclassification of explanatory variables. The second novelty has to do with the way the outcomes are evaluated. Borehole flow rates are used as a means to complement standard machine learning metrics, thus providing additional robustness to predictions. This method is demonstrated through the application of machine learning techniques to map groundwater potential across two regions of Mali. The geographical setting also represents an added value to the literature. Indeed, while there are numerous examples of GPM studies based on artificial intelligence in other continents (Naghibi et al., 2017a; Chen et al., 2018; Panahi et al., 2020), these approaches remain uncommon across Africa.*

We found another instance in the manuscript that could be misleading in terms of the objectives (section 2.5, second paragraph). We have rewritten it as: *"The rationale behind using preprocessing approaches is to rely on raw data as much as possible, instead of reclassifying it into intervals generated statistically or by expert criteria. Four scaling methods were therefore used: standardization, maximal absolute scaler (MaxAbs), maximal-minimal scaler (MaxMin) and normalization (Pedregosa et al., 2011)"*.

**[4] In the Introduction section only studies for the other continent are presented. It would be interesting to see how studies in other regions in Africa deal the groundwater potential mapping with machine learning or others methods?**

Agreed. We thank Reviewer #1 for a suggestion that allows us to highlight another important novelty of our work. GPM studies based on artificial intelligence are uncommon in Africa. To our knowledge, the only known precedent is a paper by Martinez-Santos and Renard (2019), in which the authors mapped groundwater potential in the Baoulé basin, Mali. We now note this in the last three sentences of the introduction: *"This method is demonstrated through the application of machine learning techniques to map groundwater potential across two regions of Mali. The geographical setting also represents an added value to the literature. Indeed, while there are numerous examples of GPM studies based on artificial intelligence in other continents (Naghibi et al., 2017a; Chen et al., 2018; Panahi et al., 2020, these approaches remain uncommon across Africa"*.

We also agree that there are many non-machine learning GPM studies in Africa. To acknowledge this, the second paragraph of the introduction now reads:

*"Groundwater potential mapping (GPM) is recognized as a valuable tool to underpin planning and exploration of groundwater resources (Elbeih, 2015). GPM may be understood as a means to estimate groundwater storage in a given region, as a measure of the probability of finding groundwater, or as a prediction as to where the highest borehole yields may occur (Díaz-Alcaide and Martínez-Santos, 2019). However, it consists of computing spatially distributed estimates for a target variable (groundwater*

*potential) based a set of dependent variables such as soil, lineaments, slope, geology, landforms, lithology, and drainage density. GPM often uses existing cartography, digital elevation models, aerial photographs, satellite imagery and geophysical information (Díaz-Alcaide and Martínez-Santos, 2019). Recent years have witnessed a growing interest in groundwater potential studies in Africa, largely as a result of the need to achieve the Sustainable Development Goal #6. The majority of these work with a combination of remote sensing, geographic information systems and geophysics (Delgado 2018, Adeyeye et al., 2019, Magaia et al 2018, Mpofu et al 2020, Owolabi et al 2020, Saadi et al 2021, Al-Djazouli et al. 2021), while others rely directly on the interpretation of information from borehole databases (Díaz-Alcaide et al 2017)."*

**[5] Also, the authors should be linking the issues of groundwater resources in the context of The Sustainable Development Goals (SDGs) to motive thereader at the end of the introduction.**

Agreed. We concur on the importance of groundwater to achieve the Sustainable Development Goals (SDGs). We now provide a mention to SDG 6 in the second paragraph of the introduction: "*Recent years have witnessed a growing interest in groundwater potential studies in Africa, largely as a result of the need to achieve the Sustainable Development Goal #6. The majority of these efforts use a combination of remote sensing, geographic information systems and geophysics (Delgado 2018, Adeyeye et al., 2019, Magaia et al 2018, Mpofu et al 2020, Owolabi et al 2020, Saadi et al 2021, Al-Djazouli et al. 2021), while others rely directly on the interpretation of information from borehole databases (Díaz-Alcaide et al 2017)*".

Also, we now mention the relevance of our work to SDG 6 in the second paragraph of the conclusions:

*"A crucial finding of this research is that conventional machine learning metrics (test score, area under the receiver operating characteristic curve), can be more representative of algorithm performance than of the actual field conditions. This is particularly relevant when attempting to develop spatially-distributed predictions. Double-checking algorithm results with an independent groundwater dataset (borehole flow rates in this case) is recommended to ensure that map outcomes are accurate. Machine learning approaches are thus seen as a means to underpin borehole siting initiatives at the regional scale, although it is recognized that local-scale fieldwork is needed for optimal outcomes. In the context of Sustainable Development Goal #6, predictive groundwater maps such as those developed in the course of this research may be of use to private investors willing to participate in improving water access in remote regions, as well as to government officers, cooperation funds and international donors".*

[6] **You could add this reference in the introduction section: A new method to map groundwater potential at a village scale, based on a comprehensive borehole database. An application to Sikasso, Republic of Mali by Ana Carolina Gonçalves Delgado, 2018.**

Agreed. The suggested reference is a product of our own research group. It has been added to the second paragraph of the introduction section.

**[7] Add a new section (or put some sentences) about the limitations of machine learning techniques to study groundwater potential zones.**

Agreed. We have added a limitations section (section 3.5) where we present the main limitations of our approach. This section reads:

*3.5 Limitations*

[revised manuscript text omitted]

**[8] The overfitting problem is one of the drawbacks that affect the accuracy of models in machine learning. Take into account this issue in the introduction.**

Agreed. To maintain the flow of the text, we have placed our comment on overfitting in the limitations section, rather than in the intro. Please see also our answer to [7].

[9] **The topic of validation of Groundwater Potential Map by is not mentioned. In my opinion, it is oneof the major limitations of the study. If so, this topic should be discussed in more detail.**

We agree that the original manuscript does not speak of validation explicitly. However, sections 3.2 and 3.4 present very detailed validation procedures and scores. Most of section 3.2, including Table 1, is devoted to outlining the results in terms of standard machine learning metrics (f-1, AUC, test score). These represent different takes on how well each of the models is able to "guess" an unknown outcome based on what it learned during the training process. This is the routine validation procedure of any machine learning study.

Then, in section 3.4, we go beyond standard machine learning validation metrics to compare the maps we produced with actual borehole yield data (see also Table 3). This is an additional validation procedure that is seldom carried out in groundwater potential studies. It adds robustness to our results. Please note this is also a chief conclusion of our work. The conclusions section (second paragraph) reads:

*"A crucial finding of this research is that conventional machine learning metrics (test score, area under the receiver operating characteristic curve), can be more representative of algorithm performance than of the actual field conditions. This is particularly relevant when attempting to develop spatially-distributed predictions. Double-checking algorithm results with an independent groundwater dataset (borehole flow rates in this case) is recommended to ensure that map outcomes are accurate. (…)"*

**[10] Did you limit the validation of your model with cross-validation? Or do you have the intention to integrate the external validation?**

Please see [9] above. External validation is provided by comparing machine learning validated outcomes with a different dataset (borehole flow rates).

**[11] The authors need to describe in the methodology section a sub-section of "Multicollinearity analysis" before presenting the results in section 3.1. Did you use the variance inflation factor (VIF) and tolerance (TOL) indices as are customarily used to estimate multicollinearity of predictive factors in machine learning modelling? If so, explain in the manuscript.**

Agreed. We now describe this in the second paragraph of section 2.6, which now reads:

*"Collinearity occurs when two or more variables are highly correlated. This can affect the performance of the classifiers by attributing extra weight to an input variable or by adding noise to the outcomes. Interpretability can also be impaired because the regression coefficients of certain algorithms are not uniquely determined (Martínez-Santos et al., 2021). MLMapper incorporates a collinearity analysis function to prevent collinearity from adversely affecting the results. Collinearity analysis is performed before running the algorithms. Pairwise correlation among explanatory variables is computed and correlation coefficients are expressed in a range between -1.0 (inverse correlation) and 1.0 (direct correlation). Highly-correlated explanatory variables may be excluded from the algorithm training procedure."*

As explained in the text above we used a different procedure (not VIF/TOL). Our approach is based on pairwise correlation analyses (see also Fig 8).

**[12] What is the effect of sample size on the different machine learning models for groundwater potential mapping in your research?**

We carried out many test runs. The first one included all villages (1605 human settlements). We observed that the machine learning models could not find clear associations between explanatory variables and groundwater potential. We concluded that human settlements with a single borehole might not be statistically representative, especially in cases where the average yield is low (these could represent minimum extraction flow rates determined by the type of pump, rather than by aquifer parameters). Different sample sizes were defined as per the number of boreholes in the villages. The analysis led to select villages with five or more boreholes as our optimal sample.

**[13] Did you try to make a sensitivity analysis of the effect of each factor (explanatory variables) on thegroundwater potential map, i.e., when you decide to eliminate one or more factors??**

Yes. The RFECV procedure described in section 2.6 (third and fourth paragraphs) includes a sensitivity analysis. This procedure analyzes the impact of eliminating each of the explanatory variables. Once all variables are appraised, the number of features leading to the highest test score is selected. This automatic

procedure is responsible for discarding or keeping each variable.

**[14] What is the resolution chosen to develop the thematic layers? Because the various GIS layers come with different spatial resolutions. You need to clarify this aspect.**

Agreed. As suggested, we now include the explanatory variables used in groundwater potential mapping in a Table. The information about the resolution of each explanatory variable were included in the Table below:

**Table 1.** Explanatory variables used in GPM. The scale/resolution, acquisition time and source of data for each factor are provided.

| Explanatory variables | Scale/resolution | Time (dd/mm/yyyy) | Source of data |
|---|---|---|---|
| Alteration Band Ratio | 30 meters | 07-16/03/2020 | Own elaboration from Landsat 8 |
| Clay content | 250 meters | N/A | SoilGrids250m 2.0 |
| Curvature | 30.53 meters | N/A | Own elaboration from DEM |
| Saturated thickness | 30.53 meters | N/A | Own elaboration from DEM and borehole database |
| Water table Depth | 30 meters | 2010 | Own elaboration from DNH (2010) |
| Distance from channels | 30.53 meters | N/A | Own elaboration from DEM |
| Geology | 1:5 million | N/A | British Geologycal Survey |
| Geomorphology | 30.53 meters | N/A | Own elaboration from DEM |
| Land use | 300 meters | 2009 | ESA Climate Change Initiative |
| Soil | 1:3M | N/A | European Soil Data Centre |
| Rainfall | 0.5° | 1950-2009 | CRU TS 3.21 dataset (Climatic Research Unit at the University of East Anglia) |
| Drainaige density | 30.53 meters | N/A | Own elaboration from DEM |
| Thickness matrix | 30.53 meters | N/A | Derived from DEM and borehole database |
| Elevation (DEM) | 30.53 meters | 23/09/2014 | Shuttle Radar Topography Mission (SRTM) |
| NDVI | 30 meters | 07-16/03/2020 | Own elaboration from Landsat 8 |
| NDWI | 30 meters | 07-16/03/2020 | Own elaboration from Landsat 8 |
| Slope | 30.53 meters | N/A | Own elaboration from DEM |
| SPI | 30.53 meters | N/A | Own elaboration from DEM |
| TWI | 30.53 meters | N/A | Own elaboration from DEM |

**[15] The sub-sections in Section 2 on "Material and methods" need to be reorganised for better reading. For example, you could define the title "Materials and methods": Study area; Data used (Borehole database, Explanatory variables/Thematic layers, etc); and Methods (Random Forest, AdaBoost, Gradient Boosting, Decision Tree, and Extra Trees classifiers); Tools used to process the data, etc.**

Agreed. We have rearranged this section so that:
2.1 Study area.
2.2 Data (instead of "borehole database"), including the old section 2.4 (now 2.2.1 and 2.2.2, please see [31] below)
2.3 Predictive mapping software
2.4 Preprocessing of explanatory variables

2.5 Supervised classification routine
2.6 Machine learning metrics for algorithm evaluation

**[16] A table of data sources must be put to increase the clarity and to ease readers' understanding.**

Agreed. Please see [14].

**[17] Do you have performed quality control of datasets before the modelling?**

Yes. The borehole database was verified on site by government officers (Direction Nationale de l'Hydraulique). Then they made it available for us.

**[18] Did you use all explanatory variables to map groundwater potential in Figure 10? If so, could you specify the variables used to develop the final products? Because, on Page 20 of MS, you mentioned that the outcomes show that elevation, rainfall, geology, and drainage density, among others, are the most important factors conditioning the groundwater potential. Did you use thesefour (04) explanatory variables in reality? Specify exactly the variables used in the final models.**

Agreed, the way we presented this in the manuscript could lead to confusion. Each model uses a different set of explanatory variables Each model picks its own by means of the Recursive Feature Elimination Cross Validation procedure. The second paragraph of section 3.3 now reads:

*"A major advantage of incorporating recursive feature elimination is that it eliminates part of the expert bias associated with the choice of explanatory variables. In this case, the fact that all variables are used by at least two of the best-performing algorithms suggests that the initial choice of explanatory variables was appropriate. However, feature selection reveals clear differences among the classifiers. Under standardized scaling, RFC only required three explanatory variables to predict groundwater potential (precipitation, expected saturated thickness and elevation). ABC and GBC used eight each, while ETC and DTC used eleven and fourteen, respectively. Under MaxAbs scaling, the RFC and ABC used three and four variables, respectively. GBC algorithm worked with six, DTC used sixteen and ETC seventeen. The variables used by each of the best-performing algorithms are presented in Figure 8".*

[19] **The conclusion is very general. To be check according to the revision of MS.**

Agreed. We have added the importance of this research to SDG 6 and to stakeholders in the second paragraph of the conclusions:

*"A crucial finding of this research is that conventional machine learning metrics (test score, area under the receiver operating characteristic curve), can be more representative of algorithm performance than of the actual field conditions. This is particularly relevant when attempting to develop spatially-distributed predictions. Double-checking algorithm results with an independent groundwater dataset (borehole flow rates in this case) is recommended to ensure that map outcomes are accurate. Machine learning approaches are thus seen as a means to underpin borehole siting initiatives at the regional scale, although it is recognized that local-scale fieldwork is needed for optimal outcomes. In the context of Sustainable Development Goal #6, predictive groundwater maps such as those developed in the course of this research may be of use to private investors willing to participate in improving water access in remote regions, as well as to government officers, cooperation funds and international donors."*

We have also added a paragraph where we deal with case-specific outcomes to make our conclusions more concrete (last paragraph of the conclusions).

*From a regional perspective, groundwater potential closely resembles rainfall. Despite the predominance of low-permeability basement rocks, medium to high groundwater potential observed in southern areas seems to be associated with high rainfall and well-developed weathering mantles. In contrast, good conditions for groundwater storage in the north present a low potential due to limited precipitation. The*

*central part is characterized by a medium groundwater potential in the plains, as well as by a high potential in alluvial sediments of the major river systems and low potential in the highlands .*

**[20] Limitations of the research should be addressed.**

Agreed. We have added a new section on limitations. Please see [7].

**[21]I suggested the authors separate the results and discussion. I suggest developing in the new discussion section "the model validation/performance and comparison"; "assessment of variable importance"; limitations of the research", etc.**

We thank Reviewer #1 for this suggestion. We observe that many papers in this journal present results and discussion together. Given the complexity of our manuscript, we fear that changing the structure of this section now could potentially lead us to the loss of important information. Therefore, we prefer to keep it as it is unless strictly necessary.

[22] **Furthermore,try to compare the outcomes of your research with other studies available in the literature on themapping of groundwater potential such as the GIS-based Dempster–Shafer model, etc.**

We agree that a reference to the Dempster-Schafer model was missing. It is however difficult to provide a meaningful comparison with it because we would need to apply it specifically to our study region to draw conclusions. Besides, we lack the experience with that particular model to make comparisons on a methodological level. Instead, we now mention it in the literature review and provide references. The third paragraph of the introduction now reads:

*"There are two main approaches to GPM: expert-based decision systems and machine learning methods. Expert-based systems have existed for a long time (DEP, 1993). These include multi-influence factor techniques (Magesh et al., 2012; Nasir et al., 2018; Martín-Loeches et al 2018), analytical hierarchy processes (Mohammadi-Behzad et al., 2019; Al-Djazouli et al., 2020), and Dempster-Shafer models (Mogaji and Lim 2018, Obeidavi et al 2021). Other frequently used methods are weight of evidence and frequency ratio analysis (Falah and Zeinivand, 2019; Boughariou et al., 2021). Machine learning is comparatively newer. A major difference between machine learning and these approaches is that supervised classification uses the advantages of artificial intelligence to find complex associations among explanatory variables that might otherwise pass unnoticed. Hence, machine learning is well suited to map complex spatially-distributed variables such as groundwater occurrence. Algorithms used in the GPM literature include Mixture Discriminant Analysis (Al-Fugara et al., 2020), Random Forest (Kalantar et al., 2019; Moghaddam et al., 2020), Boosted Regression Tree (Naghibi et al., 2016), Logistic Regression (Ozdemir, 2011; Chen et al., 2018; Nhu et al., 2020), Support Vector Machines (Naghibi et al., 2017b), Neural Networks (Lee et al., 2012; Panahi et al., 2020) and Ensemble methods (Naghibi et al., 2017a; Martínez-Santos and Renard, 2019; Nguyen et al., 2020b)."*

**[23] The discussion is incomplete, authors must address the uncertainty in groundwater potential mapping (deficiencies of data quality; biased and absent data, sample sizes, missing covariates, and also errors in the structural and nature of the model, etc). Add some references.**

Agreed. We have added a new section on limitations. Please see [7].

**[24] In the Conclusion section, specify the utility of this research and the potential users. For example,who could use the prepared maps?**

Agreed. We have improved the second paragraph of the conclusions. It now reads:

*A crucial finding of this research is that conventional machine learning metrics (test score, area under the receiver operating characteristic curve), can be more representative of algorithm performance than of the actual field conditions. This is particularly relevant when attempting to develop spatially-distributed predictions. Double-checking algorithm results with an independent groundwater dataset (borehole flow rates in this case) is recommended to ensure that map outcomes are accurate. Machine learning approaches are thus seen as a means to underpin borehole siting initiatives at the regional scale, although it is recognized that local-scale fieldwork is needed for optimal outcomes. In the context of Sustainable Development Goal #6, predictive groundwater maps such as those developed in the course of this research may be of use to private investors willing to participate in improving water access in remote regions, as well as to government officers, cooperation funds and international donors.*

**[25] Is it possible to improve the performance of the best machine learning models in your study? Which additional predicting variable (s) (even if such information is scarce) could be added to improve the results?**

Yes. It is definitely possible, but we would need better input data (which is currently unavailable). We explain this in the newly added section 3.5). Please see [7].

**[26] I think that Table 2 on Page 19 could be placed in a new section in Supporting Information.**

Agreed. Fixed.

**[27] Abstract section: Page 1. The abstract is very long. It should be shortened and focused. The abstract should be thoroughly revised according to the revision of the manuscript.**

Agreed. The abstract is now down from 350 to 250 words. We believe it is more focused now. It reads:

*"Groundwater is crucial for domestic supplies in the Sahel, where the strategic importance of aquifers will increase in the coming years due to climate change. Groundwater potential mapping is a valuable tool to underpin water management in the region, and hence, to improve drinking water access. This paper presents a machine learning method to map groundwater potential in two regions of Mali. A set of explanatory variables for the presence of groundwater is developed first. Scaling methods (standardization, normalization, maximum absolute value and min-max scaling) are used to avoid the pitfalls associated with the reclassification of explanatory variables. Noisy, collinear and counterproductive variables are identified and excluded from the input dataset. Twenty machine learning classifiers are then trained and tested on a large borehole database (n=3,345) in order to find meaningful correlations between the presence or absence of groundwater and the explanatory variables. Tree-based algorithms (accuracy >0.85) consistently outperformed other classifiers. Maximum absolute value and standardization proved the most efficient scaling techniques. Borehole flow rate data is used to calibrate the results beyond standard machine learning metrics, thus adding robustness to the predictions. The southern part of the study area was identified as the better groundwater prospect, which is consistent with the geological and climatic setting. Outcomes lead to three major conclusions: (1) picking the best performers out of a large number of machine learning classifiers is recommended as a good methodological practice; (2) standard machine learning metrics should be complemented with additional hydrogeological indicators whenever possible; and (3) variable scaling helps minimize expert bias".*

**[28] Keywords: I propose to delete "big data; climate change and water access"; and add "Groundwaterpotentiality, and GIS".**
Agreed. Fixed

**[29] Specific comments: Page 2. Paragraph 1. Line 2, rewrite the sentence "Today, 2.5 billion people…." by "Today, 2.5 billionpeople around the World…".**
Agreed. Fixed.

**[30] Page 2. Paragraph 3. Line 1. Introduce a sentence before this: "There are two main approaches toGPM: expert-based decision systems and machine learning methods".**
Agreed. Fixed.

**[31] Page 3. For section 2.4, Material and methods, I suggest separating them into two sections.**
**2.4.1 Definition of target**
**2.4.2.2 Explanatory variables/Thematic layers**
**In this new section, I propose to describe the explanatory variables by order according to Figure 6. Also, I propose to prepare explanatory variables used in groundwater potential mapping in a Table. Inyour Table, you could, for example, put in 4 columns (Type of data layers/ Explanatory variables;scale/resolution, time, available format, and source of data).**

Agreed. The first two paragraphs are now section 2.2.1 (Target variable). The remainder of this section is now 2.2.2 Explanatory variables.

We have prepared the table as suggested. Please see our answer to [14].

We have reworked Figure 6 so that it matches the explanation in the text in order.

**[32] Page 6. Line 3. Specify correctly if the numbers 530 and 452 are the number of villages?**

Agreed. The text is correct. Both numbers refer to the number of villages. 530 is the number of villages with a 100% success rate, of which 452 have only one borehole.

**[33] Page 9: Put in order Figure 5a; before Figure 6.**

Agreed. Fixed.

**[34] Page 9: Rewrite the last sentence by also, it was used to.**

Agreed. Fixed. It now reads *"It was also used to obtain…"*

**[35] Page 9. Figure 4 must be centered.**

Agreed. Fixed.

**[36] Page 12: Fig.12: order the number of figures following the description found on page 9, i.e.: curvature,slope, topographic wetness index (TWI),**

Agreed. Fixed.

**[37] Page 13: Number 2.5 was repeated on page 14. Check it. I have the impression that the authors did not take the time to proofread the document.**

Fixed. We did take the time to proofread the document. Minor mistakes happen.

**[38] Page 13: 2ⁿᵈ paragraph. The objectives of the study mentioned at various locations in the manuscript should be summarized at the end of the introduction (see comments above). Why did you put the maingoal of the study here? I think that the objective must be found in the introduction section.**

Agreed. Fixed. This was just a reminder of the goal. We have deleted it as explained above, so that there

is no confusion. Please see our answer to [3].

**[39] Page 14. 3rd paragraph. Did you fix the number of iterations at 500 in this study? It is the default value of model? Justify how this number was established.**

Agreed. The third paragraph in section 2.6 now reads:

*"Random-search parameter fitting increases the accuracy of the predictions by identifying the best combination of those parameters that govern each algorithm. The random search cross-validation function needs an algorithm, a scoring metric to evaluate the performance of the different hyperparameters and a dictionary with the hyperparameter names and values. In regard to the previous version of MLMapper, which used grid-search cross validation, this provides additional flexibility and reduces computational time. A sensitivity analysis of the number of iterations was performed. The best compromise between results and computational cost were obtained by fixing the number of iterations at 500. No significant improvement in scoring metrics was observed for higher values, whereas running times increased considerably."*

**[40] Page 17: you mentioned in the first line that "The AUC exceeds 0.90 in all cases". I'm not sure about this affirmation because, if you analyse Table 1, you observe that in the MaxAbs scaling method AdaBoost shows an AUC value of 0.898. Could you rewrite your sentence to take into account this case? Or maybe use AUC mean because this value exceeds in all cases.**

Agreed. The sentence now reads:
*"AUC exceeds 0.90 in all cases except for the AdaBoost algorithm (0.898) and Decision Tree algorithm (0.893)."*

**[41] Page 20. First-line (Line 415). You mentioned Naghibi and Pourghasemi (2015), the citation is incorrect because you have three authors: Seyed Amir Naghibi & Hamid Reza Pourghasemi & Barnali Dixon. At the end of this paragraph again, you mentioned (Naghibi and Pourghasemi, 2015; Nguyen et al. 2020b). Due to this error for the citation in two places, I propose to check all references.**

We believe our referencing is correct. Naghibi and Pourghasemi (2015) is included in the reference list (page 28, last-line). This reference only has two authors:

Naghibi, S. A. and Pourghasemi, H. R.: A Comparative Assessment Between Three Machine Learning Models and Their Performance Comparison by Bivariate and Multivariate Statistical Methods in Groundwater Potential Mapping, Water Resour. Manag., 29, 5217–5236, https://doi.org/10.1007/s11269-015-1114-8, 2015.

The other paper by Naghibi, Pourghasemi and Dixon (Naghibi et al 2016) is quoted in the introduction.

We have double-checked all references anyway.

**[42] Page 21: Move Figure 8 on page 21 under the section of "3.3 Importance of explanatory variables"**

Agreed. Fixed.

**[43] Page 23: I propose to add the well locations/boreholes on the two maps in Figure 8. If possible, put on these two maps: well training and well validation with different colours of points. Also, make clear your legend of Fig.8 with the classes well-defined.**
**For example:**
**(0- 0.2) Very low;**
**(0.2- 45) Low;**
**Etc.**
**Change the term "Intermediate" in the legend by "Moderate". It is most appropriate. Change**

**"Groundwater potential" to "Groundwater potential classes".**

Agreed. Fixed. We assume the reviewer refers to Figure 10 below.

[Figure]

**Figure 11**. Mapping outcomes of the agreement map for MaxAbs scaling method and Standardized scaling method. a) Training points on the MaxAbs scaling method agreement map. b) Training points on the Standardized scaling method agreement map. c)

Testing points on the MaxAbs scaling method agreement map. d) Testing points on the Standardized scaling method agreement map.

**[44] Page 21. I repeat the need to clarify my request mentioned above (see general comments). When you analyse feature importance calculated in Figure 8, you observe that some explanatory variables are not important in the models. Could you explain more how many variables did you select to produce the outcomes of Figure 10?**

Agreed. Please see our answer to [18].

**[45] Page 23. Why did you choose to classify villages in three classes based on groundwater potential, and you show the outcomes of Groundwater potential in five classes?**

Agreed. To clarify: classes are potential outcomes. There are only two (positive and negative), and groundwater potential is classified in positive and negative for each algorithm (Figure 9). However, ensembling allows for a finer classification. In Figure 10 the arithmetic mean of all five best classifiers is computed at the pixel level. This renders six possible values (0, 0.2, 0.4, 0.6, 0.8 and 1), which represent the agreement among the best classifiers for each pixel. This in turn allows for five intervals (0-0.2 Very low; 0.2-0.4 Low; 0.4 - 0.6 Moderate; 0.6 - 0.8 High; 0.8 - 1 Very high). We represent each of these agreement level as a groundwater potential outcome.

Furthermore, by dividing the villages into five classes, the sample of villages falling into high, moderate and low potentials is very small (approximately 20 points per class compared to 170 for high potentials and 70 for low potentials). It is therefore difficult to draw large conclusions from such comparatively small samples, and so we have chosen to group them in such a way that the number of villages in each category is more balanced (nearly 70).

**[46] Page 24. Could explain more why Groundwater potential classes are three in Table 3 compared to Figure 10, where we found five classes?**

Agreed. Please see [45] above

**[47] Page 15. Section 3 on Results and discussion. Please add a new sub-section on "Validation on machine learning models.**

Please see [9].

**[48] Reference section Page 27. Rewrite this reference: Direction Nationale de l'Hydraulique (Ed.): Données Hydrogeologiques et des Forages. Direction Nationale de l'Hydraulique du Mali, 2010. to the precise country name.**
Agreed. The reference now reads:

Direction Nationale de l'Hydraulique du Mali (Ed.): Données Hydrogeologiques et des Forages. Direction Nationale de l'Hydraulique. Bamako, Mali. 2010.

**[49] Page 29. Precise the link and access date of this reference: Poggio, L. and de Sousa, L.: SoilGrids250m 2.0 - Clay content, 2020.**
Agreed. The reference now reads:

Poggio, L. and de Sousa, L.: SoilGrids250m 2.0 - Clay content, https://soilgrids.org/, Access date: 15/02/2020, 2020.

**[50] Page 30.**
**Add the access date of the reference Traore, A, Z., et al.**
Agreed. The reference now reads:

Traore, A. Z., Bokar, H., Sidibe, A., Upton, K., Ó Dochartaigh, B., and Bellwood-Howard, I.: Africa Groundwater Atlas: Hydrogeology of Mali, http://earthwise.bgs.ac.uk/index.php/Hydrogeology_of_Mali, Access date: 27/10/2020, 2018.

**[51] Add the link and access date of this reference: United Nations: Resolution A/RES/64/292. UnitedNations General Assembly, United Nations, 2010.**
Agreed. The reference now reads:

United Nations: Resolution A/RES/64/292. United Nations General Assembly, United Nations, https://undocs.org/pdf?symbol=en/a/res/64/292, Access date: 10/03/2021, 2010.

**[52] Technical corrections. Page 5. Line 115-120. Add the unity of mean water depth to be coherent in the sentence, because youhave put the unity of mean electric conductivity.**
Agreed. Fixed.

**[53] Page 6: in Figure 2 B, write correctly m³/h**
Agreed. Fixed.

**[54] Page 8. Line 160-165, add the comma in (BGS, 2021). Also, on Page 9 and the title of Figure 4, add acomma to the same reference.**
Agreed. Fixed.

**[55] Page 11, line 5. "semiarid" "semi-arid"**
Agreed. Fixed.

**[56] Page 13:  Equation 6; define        $\tilde{x}$**
Agreed. Fixed.

**[57] Page 25. Delete "s" in Conclusion.**
Agreed. Fixed.

On a final note, we would like to thank Reviewer #1 again for a thorough review of our manuscript. We hope our answers will be enough to merit publication in HESS.

---

## Author Comment (AC2)

**[1] The manuscript "Preprocessing approaches in machine learning-based groundwater potential mapping: an application to the Koulikoro and Bamako regions, Mali " represents an important contribution aligned with the objective of theHESS journal and can interest the scientific community working on machine learning applied in water management. Concerning the scientific quality, I think that the used scientific approach and applied methods are interesting but the sections of the manuscript have unbalanced structure andsome sections are inappropriate and need in-depth analysis with improving the used English language. For that, I think this paper needs major modification and resubmission**

Thank you for the positive feedback. We have strived to address all comments and suggestions, as well as to incorporate them to our manuscript.

**General Comments. The introduction :**

**[2] -the section dedicated to the Reviews of literature concerning Groundwater potential mapping studies should be more developed with the presentation of the brief results of the pertinent studies.**

Agreed. We have improved the second and third paragraphs of the introduction, with an emphasis on the machine learning literature. The review of literature concerning GPM studies now reads:

*"Groundwater potential mapping (GPM) is recognized as a valuable tool to underpin planning and exploration of groundwater resources (Elbeih, 2015). GPM may be understood as a means to estimate groundwater storage in a given region, as a measure of the probability of finding groundwater, or as a prediction as to where the highest borehole yields may occur (Díaz-Alcaide and Martínez-Santos, 2019). However, it consists of computing spatially distributed estimates for a target variable (groundwater potential) based a set of dependent variables such as soil, lineaments, slope, geology, landforms, lithology, and drainage density. GPM often uses existing cartography, digital elevation models, aerial photographs, satellite imagery and geophysical information (Díaz-Alcaide and Martínez-Santos, 2019). Recent years have witnessed a growing interest in groundwater potential studies in Africa, largely as a result of the need to achieve the Sustainable Development Goal #6. The majority of these work with a combination of remote sensing, geographic information systems and geophysics (Delgado 2018, Adeyeye et al., 2019, Magaia et al 2018, Mpofu et al 2020, Owolabi et al 2020, Saadi et al 2021, Al-Djazouli et al. 2021), while others rely directly on the interpretation of information from borehole databases (Díaz-Alcaide et al 2017).*

*The literature shows that there are two main approaches to GPM, namely, expert-based decision systems and machine learning methods. Expert-based system methods have been used for a long time (DEP, 1993). These include multi-influence factor techniques (Magesh et al., 2012; Nasir et al., 2018; Martín-Loeches et al 2018), analytical hierarchy processes (Mohammadi-Behzad et al., 2019; Al-Djazouli et al., 2021), and Dempster-Shafer models (Mogaji and Lim 2018, Obeidavi et al 2021). Other frequently used methods are weight of evidence and frequency ratio analysis (Falah and Zeinivand, 2019; Boughariou et al., 2021). Machine learning is comparatively newer. A major difference between machine learning and expert approaches is that supervised classification uses the advantages of artificial intelligence to find complex associations among explanatory variables that might otherwise pass unnoticed. Hence, machine learning is well suited to map complex spatially-distributed variables such as groundwater occurrence. The GPM literature showcases a wide variety of supervised classification approaches. Thus, Al-Fugara et al. (2020) used mixed discriminant analysis to map spring potential in a watershed of Jordan; much like Odzemir (2011) mapped spring potential in a Turkish basin by means of a logistic regression method. Random forests have proved adept at mapping groundwater potential, both in mountain bedrock aquifer (Moghaddam et al., 2020), as well as in large metasedimentary basins (Martínez-Santos and Renard 2019). Other supervised classification methods include boosted regression trees (Naghibi et al., 2016), support vector machines (Naghibi et al., 2017b), neural networks (Lee et al., 2012; Panahi et al., 2020) and Ensemble methods (Naghibi et al., 2017a; Martínez-Santos and Renard, 2019; Nguyen et al., 2020b)."*

**[3] the introduction missed the presentation of the water resources problems in the study area and the need to elaborate the Groundwater potential map**

Agreed. We have incorporated this information to the first paragraph of the introduction. This now reads:

*"Water is crucial for human beings. Water provides food security, cleanliness and hydration, which translates into health, economic activity and arguably, better education opportunities (United Nations, 2002, 2010). Today, 2.5 billion people depend exclusively on groundwater for their domestic supply (Grönwall and Danert, 2020). Groundwater is particularly crucial in most of the Sahel, where rainfall and surface water are absent for several months (Llamas and Martínez-Santos, 2005; Díaz-Alcaide et al., 2017). In a context of climate change, in which rainfall is expected to decrease in most arid and semiarid regions and drought episodes are likely to become more intense (Arneth et al., 2019), groundwater resources will be increasingly relied upon. This is the case of the Republic of Mali, where access to drinking water and sanitation remains a concern for a large part of the population. In 2017, only 68% of the rural population had "at least basic" drinking water access, while 24% still relied on unimproved water sources (UNICEF/WHO, 2019). Since the country's aquifers are still relatively unknown, there is an impending need to endow water managers with tools to optimize groundwater use."*

**[4] Then the results discussed must be more in-depth, especially by explaining the results of the GPM obtained in connection with the hydrogeological context of the study area and theused explanatory parameters.**

Agreed. We have rewritten the first paragraph of section 3.4 to comply with this observation. It now reads:

*"Classifier outcomes were extrapolated to produce groundwater potential maps. Figure 10 shows the groundwater potential predictions rendered by each of the five best-performing algorithms (Decision Tree (DTC), Random Forest (RFC), AdaBoost Classifier (ABC), Gradient Boosting (GBC) and ExtraTrees (ETC)) under the two most effective scaling methods (MaxAbs scaling method and standardized scaling method). Red areas are those in which the algorithms have found a combination of explanatory variables leading to a negative potential. In turn, green zones represent a positive groundwater potential. GPM outcomes show a gradient characterized by the predominance of high potential areas in the south to a greater proportion of negative areas in the north. This appears to be related to rainfall patterns. Low potential areas occur around mountain outcrops of the southwest. The large green zone around the southern region corresponds to the weathering mantle of basement rocks. Groundwater in basement aquifers is most often found in weathered formations and piedmonts of the outcrops. Piedmonts may exhibit high GPM because these are essentially a mixture of weathered and transported materials (Martín-Loeches et al., 2018). This area presents smooth orography, so that high GPM is mainly determined by the weathered mantle. Previous research by Diaz-Alcaide et al. (2017) attributes a medium GPM for the southern part of the Koulikoro region. Discrepancies between this and our results likely stem from the fact that they used a regional approach. Furthermore, they relied on borehole yield data, while this work only classifies groundwater potential in positive-negative terms.*

*The central part of the agreement map shows a high potential for both methods, except in the higher altitude areas. This region, consisting of consolidated metasedimentary materials, has an average aquifer thickness of 30 to 50 meters (Traore et al., 2018). High yields are associated with the weathered mantle developed in the upper part instead than with the predominant lithology. This becomes evident in the metasedimentary outcrops located in the highlands. These present a low groundwater potential because fracturing facilitates rapid groundwater percolation into the plains, where it accumulates in the alteration zone. Areas near major rivers, such as the Niger, Sankarani and Bani, also have a high potential. This is attributed to the high permeability of alluvial sediments. The northern part of the study area, formed by consolidated and unconsolidated sedimentary materials, has a low groundwater potential. Although geological conditions are more favorable for groundwater due to the type of materials than in other areas of the region groundwater potential is limited by low rainfall. This is demonstrated by the feature importance analysis, which shows that precipitation is one of the two most important explanatory variables for groundwater occurrence."*

**[5] The methodology. (1) The hydrogeological context of the studies area is unfairly presented; (2) then the explanatory parameters used are unclearly presented. It is important to explainin-depth these used data to enrich the explanation of the results of GPM.**

Agreed. We improved section 2.1 to provide a better hydrogeological background. It now reads:

*"Figure 2 shows the major geological domains of the study area (BGS, 2021). The rocks that make up the Precambrian craton (south) are composed mainly of gneiss, schist and quartzite, representing metamorphosed volcanic-sedimentary sequences. The original sedimentary layers, which include shale, arkose, gravel and conglomerate, were intercalated with volcanic rocks, such as basalt, gabbro, dolerite, rhyolite and tuff. Further north, metasedimentary rocks of Proterozoic age, predominantly low-medium grade metamorphosed sandstones, with varying amounts of mudstone and limestone, take up over 50% of the study area. Volcanic outcrops (basalts and gabbros) are located in the central sector and in the northern end. Sedimentary rocks (sandstone, limestone and shale) of Cambrian-Carboniferous age and Cretaceous-Tertiary age occur in the northern third of the study area. Quaternary fluvial deposits associated with the Niger River are observed along the riverbed (Traore et al., 2018).*

[Figure]

**Figure 2**. *Geological map with the main units that outcrops in the study area (adapted from BGS, 2021)*

*From a hydrogeological perspective, four major aquifer units are distinguished (Traore et al., 2018). These include basement aquifers, aquifers linked to fractures and intergranular porosity of consolidated sedimentary rocks (Precambrian and Paleozoic), aquifers formed in intrusive volcanic rocks, and aquifers in unconsolidated sedimentary materials (Fig. 3).*

*Basement aquifers are mostly located towards the south of the Koulikoro region. These are characterized by a thick weathered mantle. The average thickness of the weathered formation over the basement in this region is between 10 and 50 meters. In these aquifers, groundwater flows preferentially in the weathered mantle, and, within this, the lower part is generally more transmissive due to lower clay content. The upper part is less permeable to flow but can still be important as a groundwater reservoir. Fractures can increase reservoir permeabilitym although their storage capacity is typically low (Martín-Loeches et al., 2018). Borehole yields range from 4 to 6 m³/hour (Traore et al., 2018).*

*The Precambrian metasedimentary materials are located in the central part of the Koulikoro region. Metasediments are considered a mixed permeability aquifer: low permeability layers provide higher storage, while more fractured layers present higher permeability and lower storage. Mean aquifer thickness ranges from 30 to 50 meters and the average yield varies from 5 to 10 m³/hour. However, some boreholes yield exceeds 100 m3/hour. In the north, the fractured Paleozoic rocks allow water to flow through the sandstone and limestone layers. Average borehole yields are around 6 m³/hour and the fractured horizons are about 40-45 m thick. Finally, unconsolidated sedimentary materials are composed by shales and argillaceous sandstone interbedded with limestone. The average borehole yields around 7 m³/hour. The thickness of the saturated zone ranges from less than 100 m to over 400 m (Traore et al., 2018)."*

*Furthermore, we now add a figure showcasing the region's major hydrogeological domains (below).*

[Figure]

**Figure 3**. Main aquifer units of the study area (Adapted from Traore et al. (2018))

**[6] (2) Then the explanatory parameters used are unclearly presented. It is important to explainin-depth these used data to enrich the explanation of the results of GPM.**

Agreed. We have reorganized the explanatory variables section in two subsections. The first one deals specifically with the target variable, whereas the second is devoted to the explanatory variables. This now reads.

*2.2.1 Target variable*

[revised manuscript text omitted]

QGIS 3.0's Geomorphon plugin (Jasiewicz and Stepinski, 2013) was used to prepare the landform map. This approach uses DEM for the classification and mapping of landform features based on the principle of pattern recognition, rather than on differential geometry. By default the Geomorphon plugin classifies landforms in ten different categories. Because some of them can be expected to play a similar role in the context of GPM, these were subsequently regrouped in four (Fig. 5a).

Soil descriptions (Fig. 5b) of the study area were obtained from the European Soil Data Centre (Dewitte et al., 2013). About 45% of the region is characterized by the presence of Pisoplinthic Plinthosols a type of soils with plinthite (Fe-rich), strongly cemented to indurated concretions or nodules, humus-poor mixture of kaolinitic clay and other products of strong weathering (IUSS Working Group, 2015). It usually changes irreversibly to a layer with hard concretions or nodules or to a hardpan on exposure to repeated wetting and drying. Hypoluvic arenosols are present in 20% of the total surface and are characterized by being deep sandy soils, which explains their generally high permeability. These are residual sandy soils following in situ weathering of rocks generally rich in quartz. Nearly 13% of the study are characterized by Petric Plinthosols, that share multiple features with Pisoplinthic Plinthosols, which share multiple characteristics with Pisoplintic Plinthosols but unlike the latter are arranged in continuous or fractured sheets of connected concretions or nodules and are strongly cemented to indurated. Lithic Leptosols (very thin soils on continuous rock and extremely rich in coarse fragments with continuous rock from ≤ 10 cm from the soil surface), Haplic Lixisols (higher clay content in the subsoil than in the topsoil, as a result of pedogenetic processes), Eutric Regosols (very weakly developed mineral soils in unconsolidated materials) constitute about 15% of the study area.

The study area is clearly divided in terms of land use (Fig. 5c). There seems to be a clear association with the precipitation gradient. The southern part is characterized by open broadleaf deciduous forest (ESA, 2010). The central part is characterized by an alternation of shrublands, mosaics of cropland vegetation and rainfed cropland. West of Bamako, in the sparsely populated mountains, there are forests mixed with shrublands. The northern part of the study areais dominated by cropland mosaics and, further north, the landscape is made up of open grasslands, sparse vegetation and bare areas.

Boreholes in the study area are often drilled until the unaltered bedrock is reached. As a result, borehole depth can be a suitable proxy for aquifer thickness (Fig. 5d). Because the borehole database includes static level measurements, an expected saturated thickness layer was computed by subtracting one from

*the other (Fig. 5d).*

[Figure]

**Figure 5.** *Explanatory variables used to predict the GPM: a) geomorphology b) Land use (A.a = artificial areas; W.b = water bodies; M.c.v = Mosaic cropland vegetation; R,c = Rainfed cropland; B.a = Bare areas; O.g = Open grassland; M.v./c = Mosaic vegetation/cropland; C./O.s = Close to open shrubland; B.e or s.f. and M.f/s = Broadleaved evergreen or semidecidous forest and Mosaic forest / shrubland) c) Soil (Eu.Cam. = Eutric Cambisols; Eu.Nit. = Eutric Nitrisols; Eu.Reg. = Eutric Regosols; Hap.Lix. = Haplic Lixisols; Hap.Ver. = Haplic Vertisols; Hyp.Are. = Hypoluvic Arenosols; Lit.Lept. = Lithic Leptosols; Pet.Pl. =Petric Plinthosols; Pis.Pl. = Pisoplinthic Plithosols; Und.Gl. = Undifferentiated Gleysols; Und.Fl. = Undifferentiated Fluvisols; Vet.Cam = Vetric Cambisols) d) expected thickness matrix.*

Another important variable in terms of aquifer recharge is precipitation, as both can be assumed to be correlated to some extent. Rainfall data in this case represents the mean annual precipitation for the 1950-2009 interval (Fig. 6a)

Satellite monitoring does not penetrate deep into the ground, but provides information about features that may be associated with shallow groundwater (Díaz-Alcaide and Martínez-Santos, 2019). This can be important in the case at hand, where the borehole database shows the static level to remain around 5-15 m below the surface (Fig. 6b). Vegetation-related indices can be useful in this context, particularly when computed at the end of the dry season (Fig. 6c,d). Take for instance the normalized difference vegetation index (NDVI, Fig. 6c), which is an estimate of vegetation vigour and is derived from the response of vegetation to red and visible infrared wavelengths (Xie et al., 2008). Similarly, the normalized difference water index (NDWI, Fig. 6d) is used as a measure of the amount of water in the vegetation or soil moisture (Xu, 2006). Based on Landsat 8 products, the NDVI and the NDWI are computed as per Eq. 1 and Eq. 2, respectively, where B3 represents the green band (0.53 - 0.59 μm), B4 is the red band (0.64 - 0.67 μm) and B5 is the near infrared band (0.85 - 0.88 μm).

$$NDVI\ (Landsat\ 8) = (B5 - B4) / (B5 + B4) \qquad (1)$$
$$NDWI\ (Landsat\ 8) = (B3 - B5) / (B3 + B5) \qquad (2)$$

The digital elevation model (DEM) was obtained from the radar-based Shuttle Radar Topography Mission (SRTM), with a resolution of 1 arcsecond (30 m). The topography is a relevant factor in groundwater distribution, storage, and flow, as well as surface runoff and infiltration, are partially constrained by surface features and parameterized by properties that can be extracted from the surface data (Elbeih, 2015). In this case, the DEM was used to develop the curvature (Fig. 6e), slope (Fig. 6f), topographic wetness index (TWI, Fig. 6g), stream power index (SPI, Fig. 6h) and geomorphology layers (Fig. 5a). It was also used to obtain the channel network, which is used in turn to elaborate a drainage density map (Fig. 6i). and distance from channels layer (Fig. 6j).

The Topographic Wetness Index (TWI) represents the ease with which water may accumulate at the surface (Beven and Kirkby, 1979). Similarly, the Stream Power Index provides a measure of the erosive power of flowing water (Moore et al., 1991).

The channels extracted from the DEM are used to develop the drainage density and distance from channels maps. Drainage density is computed as the total length of the streams per catchment unit area. Distance from channels was developed by extracting all major channels into a separate layer and developing 500, 1500 and 2500 meter buffers.

Clay content in the first few meters of the surface largely determines the percolation of water into the aquifer. Therefore, an additional clay content layer (g/kg) in the top two meters of the terrain was considered (Poggio and de Sousa, 2020). This layer is obtained by state-of-the-art machine learning methods that use global soil profile information and covariate data to model the spatial distribution of soil properties around the world. (Fig. 6k). This layer provides information about subsurface clay content. To complement the information on clay content on the subsurface, an additional layer has been developed by combining bands 6 (short-wave infrared 1) and 7 (short-wave infrared 2) of Landsat 8 (Ourhzif et al., 2019). This layer provides information on clay content on the surface and the relationship with infiltration (Fig. 6l).

[Figure]

***Figure 6.*** *Explanatory variables used to predict the GPM: a) rainfall (mm/year) b) water table depth (metersc) normalized difference vegetation index (NDVI) e) curvature f) slope (degree) g) topographic wetness index (TWI) h) stream power index (SPI) i) drainage density j) Distance from channels k) Clay content (g/kg) l) alteration band ratio (B6/B7)*

**Revision suggestions:**

**ABSTRACT:**

**[7] line9: "Groundwater is crucial for domestic supplies in the Sahel"**

**it is necessary to precise the location. which Sahel?**

In our view, the dependence on groundwater for rural groundwater supply is similar across the Sahel belt, so this statement is suitably general. The specific area of the research (Mali) is mentioned a couple of lines below.

**[8] Line11 & 12: "This paper presents a machine learning method to map groundwaterpotential and illustrates it through an application to two regions of Mali". It is poorly structured sentences!**

Agreed. The abstract has been rewritten. It now reads:

*"Groundwater is crucial for domestic supplies in the Sahel, where the strategic importance of aquifers will increase in the coming years due to climate change. Groundwater potential mapping is a valuable tool to underpin water management in the region, and hence, to improve drinking water access. This paper presents a machine learning method to map groundwater potential in two regions of Mali. A set of explanatory variables for the presence of groundwater is developed first. Scaling methods (standardization, normalization, maximum absolute value and min-max scaling) are used to avoid the pitfalls associated with the reclassification of explanatory variables. Noisy, collinear and counterproductive variables are identified and excluded from the input dataset. Twenty machine learning classifiers are then trained and tested on a large borehole database (n=3,345) in order to find meaningful correlations between the presence or absence of groundwater and the explanatory variables. Tree-based algorithms (accuracy >0.85) consistently outperformed other classifiers. Maximum absolute value and standardization proved the most efficient scaling techniques. Borehole flow rate data is used to calibrate the results beyond standard machine learning metrics, thus adding robustness to the predictions. The southern part of the study area was identified as the better groundwater prospect, which is consistent with the geological and climatic setting. Outcomes lead to three major conclusions: (1) picking the best performers out of a large number of machine learning classifiers is recommended as a good methodological practice; (2) standard machine learning metrics should be complemented with additional hydrogeological indicators whenever possible; and (3) variable scaling helps minimize expert bias".*

**[9] Line 13: "A set of explanatory variables for the presence of groundwater is developedfirst"**
**I suggest to replacing the presence of groundwater by groundwater occurrence**

Agreed. Fixed. Please see [8].

**[10] Line17: "This process identifies noisy, collinear and counterproductive variables andexcludes them from the input dataset":**
**It is a result details, I suggest deleting this sentence.**

On the contrary, this is a crucial methodological detail. This step is relatively often overlooked in machine learning studies, although it is potentially important. We prefer to keep it, although we have rewritten the sentence.

**[11] Line 18, 19 & 20: "Tree-based algorithms, including the AdaBoost, Gradient Boosting, Random Forest, Decision Tree and Extra Trees classifiers were found to outperform other algorithms on a**

consistent basis (accuracy >0.85), whereas maximum absolute value andstandardization proved the most efficient methods to scale explanatory variables".

I suggest replacing by:

The results shows that the Tree-based algorithms, including the AdaBoost, Gradient Boosting, Random Forest, Decision Tree and Extra Trees classifierswere found to outperform other algorithms on a consistent basis (accuracy
>0.85), whereas maximum absolute value and standardization proved the mostefficient methods to scale explanatory variables.

Agreed. Fixed. Please see [8].

[12] Line 22 & 23: "From a methodological standpoint, the outcomes lead to three major conclusions":
I suggest replacing by: The outcomes of this study lead to three majorconclusions

Agreed. Fixed. Please see [8].

**Introduction**

[13] Line 38 & 39: "Groundwater potential mapping (GPM) is recognized as a valuable tool to underpin planning and development of groundwater resources (Elbeih, 2015)".
I suggest replacing by

Groundwater potential mapping (GPM) is recognized as a valuable tool to underpin planning and exploration of groundwater resources (Elbeih, 2015).

Agreed. Fixed.

[14] Line 41 & 42: "In practice, however, it consists in computing spatially-distributedestimates for a target variable (groundwater potential) based a set of explanatoryvariables"
What are the explanatory variables, you should explain them, I suggest toreplace these sentences by:

However, it consists of computing spatially distributed estimates for a target variable (groundwater potential) based a set of dependent variables such as soil,lineaments, slope, geology, landforms, lithology, and drainage density (Díaz-Alcaide and Martínez-Santos 2019a)

Agreed. Fixed.

[15] Line 42, 43 & 44: "GPM typically relies on existing cartography, digital elevation models obtained from satellite, aerial photographs, satellite imagery and geophysical information (Schetselaar et al., 2007)".
The GPM based on the assembling of data from different sources. I suggestreplacing by:

GPM typically relies on the compilation of data derived from existing maps, aerialphotographs, satellite imagery, and airborne geophysical information (Schetselaar et al. 2008).

Agreed. Fixed.

[16] Line 46: "There are two main approaches to GPM: expert-based decision systems andmachine learning methods".
I suggest replacing by:

Recently, expert-based decision systems and machine learning methods havebeen implemented in many groundwater studies.
Agreed. Fixed.

**[17] Line 46 &47: "Expert-based systems have existed for a long time (DEP, 1993)"**
**I suggest replacing by: Expert-based system methods have been used for a long time(DEP, 1993)**

Agreed. Fixed.

**[18] Line 52 & 53: Algorithms used in the GPM literature include Mixture DiscriminantAnalysis (Al-Fugara et al., 2020), Random Forest (Kalantar et al., 2019; Moghaddam et al., 2020),**
**I suggest replacing by:**
**In literature, The Machine Learning Algorithms used in the GPMstudies include Mixture Discriminant Analysis (Al-Fugara et al., 2020), Random Forest (Kalantar et al., 2019; Moghaddam et al., 2020),**

Agreed, but this part of the paragraph has changed at the suggestion of the other reviewer.

**[19] Line 58: "GPM works under the assumption that the presence of groundwatercan be partially inferred from surface features"**
**I suggest replacing by:**

**The GPM is based on a common assumption is that the groundwater occurrence can be partially inferred from surface features.**

Agreed. Fixed.

**[20] Line 60 & 61: Supervised classification algorithms are trained to find the associationsbetween these variables and known groundwater data.**
**The data are trained using the algorithm not the algorithms are trained: Isuggest replacing by:**

**These explanatory variables are trained using Supervised classification algorithms to findthe associations between them and known groundwater data.**

Agreed. Fixed.

**[21] Line 64 & 65: add reference.**
**Because the number of available boreholes to train and test the algorithms is usually "small", and because the number of explanatory variables can be relatively high, a crucial issue in machine-learning studies is how explanatory variables should be reclassified in order to minimize noise**

Agreed. The sentence now reads:

"Because the number of available boreholes to train and test the algorithms is usually "small", and because the number of explanatory variables can be relatively high, a crucial issue in machine-learning studies is how explanatory variables should be reclassified or grouped in order to minimize noise and decrease the variability of the values of each conditioning factor. The technique of grouping the values of the explanatory variables is widely used. (Gnanachandrasamy et al., 2018; Qadir et al., 2020; Saravanan et al., 2020)

**[22] Line 68: add reference**
**Sometimes the intervals are based directly on expert criteria, which means that a bias may be incorporated from the beginning of the process.**

Agreed.

Sometimes the intervals are based directly on expert criteria, which means that a bias may be incorporated from the beginning of the process (Martínez-Santos and Renard, 2020)

**[23] Line 71 & 72: The outcomes of machine learning GPM studies are almost invariablyassessed by means of standard big data metrics such as precision, recall, and area under the receiver operating characteristic curve.**
**I suggest replacing by:**

**The outcomes of GPM studies using machine learning algorithms are almost invariablyassessed by means of standard big data metrics such as…. And add reference to this observation**

Agreed. Fixed. The sentence now reads:

The outcomes of GPM studies using machine learning algorithms are almost invariably assessed by means of standard big data metrics such as precision, recall, and area under the receiver operating characteristic curve (Pradhan, 2013; Naghibi et al., 2016; Chen et al., 2019).

**[24] Line 76 to Line 80: Within this context, this research presents two main additions to the literature. In the first place, it explores different scaling methods. The goal is to avoid the pitfalls associated with the reclassification of explanatory variables. Scaling is thus advocated as an essential part of algorithm training since each subsequent procedure depends on the choice of unit for eachfeature (Huang et al., 2015). Furthermore, scaling is expected to transform feature values based on a defined rule, so that all scaled features have the same degree of influence (Angelis and Stamelos, 2000).**

**I suggest replacing by:**

**Within this context, this research presents two main additions to the literature. In the firstplace, it explores different scaling methods to avoid the pitfalls associated with the reclassification of explanatory variables. Scaling is thus advocated as an essential part of algorithm training, since each subsequentprocedure depends on the choice of unit for each feature (Huang et al., 2015). Furthermore, scaling is expected to transform featurevalues based on a defined rule, so that all scaled features have the same degree of influence (Angelis and Stamelos, 2000). (Thisis d detail of methodology I propose to add to the methodology section)**

Agreed. Fixed. We have moved this paragraph to the methodology section.

**Material and methods. Study area**

**[25] Line 93 to 111: I suggest to add a hydrogeological section or a geologic map to highlightthe aquifers units of the study area**

Agreed. Please see [5] above.

**[26] Line 89 to 101: "Water in these aquifers is preferentially located in theweathered mantle, and, within this, the lower part is generally more transmissive due to lower clay content. The upper part is less permeable to flow,but can still be important as a groundwater reservoir. Fractures can produce significant quantities of water, although their storage capacity is typically low (Martín-Loeches et al.,2018)"**
**I suggest replacing by:**

**In these aquifers, groundwater flows preferentially in the weathered mantle, and, within this, the lower part is generally more transmissive due to lower clay content. The upper part is less permeable to flow but can still be important as a groundwater reservoir wherethe fractures can raise the reservoir permeability although their storage capacity is typically low (Martín-Loeches et al.,2018).**

Agreed. This sentence has been rewritten differently based on a comment by the other reviewer.

**[27] Line 107: "Some boreholes however exceed 100 m3/hour"**
**I suggest replacing by:**

**However, some boreholes yield exceeds 100 m3/hour**

Agreed. Fixed.

**[28] Line 107 & 108: "The Paleozoic rocks located in the north are determined by fractures that allow water to flow through the sandstone and limestone layers".**
**I suggest replacing by:**

**In the north, the fractured Paleozoic rocks allow water to flow through the sandstone andlimestone layers.**

Agreed. Fixed.

**Borehole database**

**[29] Line 115:  Borehole data were provided by Direction Nationale de l'Hydraulique (2010)**
**I suggest replacing by:**

**Borehole data were provided by the National Water Directorate (DNH, 2010)**
Agreed. Fixed.

**[30] Line 115 to 116: "The database contains 115 information on 5,387 boreholes**
**(3,772 successful and 1,615 unsuccessful), distributed across 1,605 humansettlements".**
**I suggest replacing by:**

**The database contains information of 5,387 boreholes (3,772 successful and 1,615unsuccessful), distributed across 1,605 fields.**

Agreed. Fixed.

**[31] Line 121 to 123: "This can be assumed to be the thickness of the (Courtois et al.,2010). Water table depth**
**I suggest replacing by:**

**There is a considerable number of boreholes with a 100% success rate (530), manyvillages are supplied by a single  borehole**

There seems to be something missing about this comment.

**[32] Line 126 to 127: For algorithm training purposes, this raises the question as towhether villages with a small number of boreholes are statistically representative, particularly in cases where the mean yield is low**
**I suggest replacing by:**

**This raises the question in the application of algorithm in the choice of training datasets,especially to whether villages with a small number of boreholes are statistically representative, particularly in cases where the mean yield is low**

Agreed. Fixed.

**[33] Line 145: Figure 3: correct the word classification metrics**

Agreed. Fixed.

[Figure]

*Figure 7*. Conceptual model of the predictive mapping procedure with MLMapper v2.0.

**[34] Line 156 to 157: Sixteen explanatory variables were selected based on an extensive review of the GPM literature (Díaz-Alcaide and Martínez-Santos 2019).**

**I think to explain in detail this extensive review in the Introduction part**

Agreed. The sentence was incorrectly worded. The extensive review was carried out by Díaz-Alcaide and Martínez-Santos (2019). The sentence now reads:

Nineteen explanatory variables were selected from an extensive review of the literature on GPM conducted by Diaz-Alcaide and Martinez-Santos (2019).

**[35] Line 161: you should add a description of the main factors that can influence thegroundwater recharge before explaining the description of each used  variables or factors in the groundwater potential mapping**

Agreed. Please see [6] (first two paragraphs).
 **[36] Line 162: Geology constrains the presence of groundwater to an importantextent**
**I think to delete this sentence**

Agreed. Fixed.

**[37] Line 173: Soils are important in GPM because soil characteristics such aspermeability…**
**I suggest replacing by:**

**Soil is important factor to determine the groundwater occurrence ……….**

Agreed. Fixed.

**[38] Line 174: Soil descriptions of the study area were obtained from the EuropeanSoil Data Centre**
**You should describe the main soils of the study area types and their characteristics**

Agreed. Please see [6]

**[39] Line 175 and 176: Integration of land use and land cover is often used in groundwaterpotential mapping studies because human activities alter hydrological dynamics (Díaz-Alcaide and Martínez-Santos, 2019).**
**I suggest replacing by:**

**Integration of land use and land cover is often used in groundwater potential mapping studies because Land use changes, which are mostly induced by human activities, affecthydrological dynamics (Díaz-Alcaide and Martínez-Santos, 2019).**

Agreed. Fixed.

**[40] Line 175 to 180: you should describe the land use of your study area and thedata used for the elaboration of this map**
Agreed. We have described the land use and now it reads:

"The study area is clearly divided in terms of land use mainly due to the north-south precipitation gradient. The land use map provided by ESA Climate Change Initiative shows that the southern part is characterized by open broadleaf deciduous forest. The central part is characterized by an alternation of shrublands, mosaics of cropland vegetation and rainfed cropland. West of Bamako, in the sparsely populated mountains, there are forests mixed with shrublands. The northern part of the study area, with less rainfall, is dominated by cropland mosaics and, further north, the landscape is made up of open grasslands, sparse vegetation and bare areas."

**[41] Line 182: You should add the reference of used rainfall data**
Agreed. Fixed. Suggested by the other reviewer, we added a table containing the reference and source of all data. Rainfall source data was CRU TS 3.21 dataset (Climatic Research Unit at the University of East Anglia).

| Explanatory variables | Scale/resolution | Time (dd/mm/yyyy) | Source of data |
|---|---|---|---|
| Alteration Band Ratio | 30 meters | 07-16/03/2020 | Own elaboration from Landsat 8 |
| Clay content | 250 meters | N/A | SoilGrids250m 2.0 |
| Curvature | 30.53 meters | N/A | Own elaboration from DEM |
| Saturated thickness | 30.53 meters | N/A | Own elaboration from DEM and borehole database |
| Water table Depth | 30 meters | 2010 | Own elaboration from Borehole database |
| Distance from channels | 30.53 meters | N/A | Own elaboration from DEM |
| Geology | 1:5 million | N/A | British Geologycal Survey |
| Geomorphology | 30.53 meters | N/A | Own elaboration from DEM |
| Land use | 300 meters | 2009 | ESA Climate Change Initiative |
| Soil | 1:3M | N/A | European Soil Data Centre |
| Rainfall | 0.5° | 1950-2009 | CRU TS 3.21 dataset (Climatic Research Unit at the University of East Anglia) |
| Drainaige density | 30.53 meters | N/A | Own elaboration from DEM |
| Thickness matrix | 30.53 meters | N/A | Derived from DEM and borehole database |
| Elevation (DEM) | 30.53 meters | 23/09/2014 | Shuttle Radar Topography Mission (SRTM) |
| NDVI | 30 meters | 07-16/03/2020 | Own elaboration from Landsat 8 |
| NDWI | 30 meters | 07-16/03/2020 | Own elaboration from Landsat 8 |
| Slope | 30.53 meters | N/A | Own elaboration from DEM |

| | | | |
|---|---|---|---|
| SPI | 30.53 meters | N/A | Own elaboration from DEM |
| TWI | 30.53 meters | N/A | Own elaboration from DEM |

**[42] Line 184: Figure 4: you should add the lineaments and faults in the geologicalmap**

Agreed. Unfortunately, there is no such map. The available geologic maps are large scale and do not contain the lineaments and faults. We attempted to extract the lineaments automatically from the DEM, but the large area of the study region led to unsuccessful results.

**[43] Line 191 & 192: DEMs are relevant because shallow groundwater flow and infiltration are partially conditioned by surface features and parameterized by properties that can be extracted from the surface data (Elbeih, 2015)**
**I suggest replacing by:**

**The topography is a relevant factor in groundwater distribution, storage, and flow, as well as surface runoff and infiltration are partially conditioned by surface features and parameterized by properties that can be extracted from thesurface data (Elbeih, 2015)**

Agreed. Fixed.

**[44] Line 197: The topographic wetness index**
**I suggest replacing by:**

**The Topographic Wetness Index (TWI)**

Agreed. Fixed.

**[45] Line 243: Figure 6. Explanatory variables used to predict the GPM: a) water table depth (meters) b) slope (degree) c) curvature d) borehole yield (m3/h) e) normalized difference vegetation index (NDVI) f) normalized difference waterindex (NDWI) g) alteration band ratio (B6/B7) h) Drainage density i) Stream power index (SPI) j) topographic wetness index (TWI) k) Claycontent 245 (g/kg) l) rainfall (mm/year)**

**What is the difference of the figure 6g (alteration band ratio (B6/B7) ) and the figure 6k (Clay content); in the text it means the same information line 230 to 233: This layer provides information on clay content on the surface and the relationship with infiltration. Clay content on the surface is calculated as per Eq. 5, where B6 is theshort-wave infrared 1 and B7 the short-wave infrared 2.**

Agreed. The text might be slightly confusing. Clay content layer was obtained by state-of-the-art machine learning methods that use global soil profile information and covariate data to model the spatial distribution of soil properties around the world. This layer provide information about the clay content in the top two meters of soil, i.e. information about subsurface clay content and was obtained from SoilGrids (Poggio and de Sousa, 2020). In contrast, the alteration band ratio was used in this study for its ability to map clay minerals using bands 6 and 7 of Landsat 8 (Ourhzif et al., 2019). This layer provides information about the surface clay content.

If our manuscript continues in the revision process, we will rewrite the information about these layers in the explanatory variables section.

**[46] Line 267: reference of equation 6**

Agreed. Fixed.

**[47] Line 273: reference of equation 7**

Agreed. Fixed.

**[48] Line 380 to 400: I find this paragraph should be added to the introductionsection to explain the use of used algorithms in literature**

Agreed. If our manuscript continues in the revision process, we will add this paragraph to the introduction section.

**[49] Line 437: Classifier outcomes were extrapolated to produce groundwaterpotential maps**
**What you want to say it is not clear!**

Agreed, the text might be slightly confusing. We refer to the fact that the algorithms were trained and validated with the information from the borehole database. In contrast, to develop the final groundwater potential maps, it's necessary to know the distribution of the explanatory variables throughout the study area. Therefore, we discussed extrapolating the patterns learned from the borehole database to the entire study area to produce the groundwater potential maps.

**[50] Line 437 to 438: Figure 9 shows the groundwater potential predictions renderedby each of the five best-performing algorithms under the two most effective scaling methods**

**I suggest adding the abbreviations of used algorithms and scaling methods betweenparentheses**

Agreed. Fixed.

**[51] Line 447: The agreement map (Fig. 10) allows for an analysis of discrepanciesamong the best performing algorithms.**
**What you want to say about the agreement map!**

Agreed. The description regarding the source of this map was in the supervised classification routine subsection:

"Each algorithm operates differently and relies on a different combination of explanatory variables, which inevitably leads to discrepancies in the predictions. In order to analyse the degree of agreement between the classifiers, an ensemble map is developed by computing the arithmetic mean at the pixel scale of those algorithms exceeding 0.85 predictive accuracy. Green pixels mean that all the best-performing algorithms agreed on a positive groundwater potential outcome (arithmetic mean = 1). Conversely, red zones represent those pixels where all the best-performing algorithms agreed on a negative groundwater potential (arithmetic mean = 0). Intermediate colours represent various degrees of agreement among the algorithms."

**[52] Line 455: Figure 9. Mapping outcomes of the top five supervised classification algorithmsfor the two best performing scaling methods. At the top the MaxAbs scaling method, below it the standardized scaling method. From left to right: AdaBoost classifier, GradientBoosting classifier, Random Forest classifier, Decision Tree classifier and Extra Trees classifier.**
**I suggest to add number or letter for each map like:**
**AdaBoost classifier, (b) Gradient Boosting classifier, (c) Random Forest classifier, (d)Decision Tree classifier and (e) Extra Trees classifier.**
Agreed. Fixed.

**Individual maps**

**Groundwater potential**

■ 0 - Negative
■ 1 - Positive

**MaxAbs scaling method**

[Figure]

**Standardized scaling method**

**Figure 10.** Mapping outcomes of the top five supervised classification algorithms for the two best performing scaling methods. At the top the MaxAbs scaling method: A) AdaBoost classifier B) Gradient Boosting classifier C) Random Forest classifier D) Decision Tree classifier E) Extra Trees classifier. below it the standardized scaling method: F) AdaBoost classifier; G) Gradient Boosting classifier H) Random Forest classifier I) Decision Tree classifier J) Extra Trees classifier.

**[53] Line 492 to 494: "On a final note, the literature features few examples of groundwater potential studies in the study area. Perhaps the only systematic precedent is the one carried out by Díaz-Alcaide et al. (2017). These authors performed a national-scale assessment of groundwater potential for the Republicof Mali based on the same borehole database that has been used in this research".**

**This is a literature review about similar studies in pilot area, I suggest to add in the Introduction section**
In this case we attempt to place our results in the context of other studies of the same area, which is suitable for the discussion.

On a final note, we would like to thank Reviewer #2 again for a thorough review of our manuscript. We hope our answers will be enough to merit publication in HESS.

---

## Author Response (AR1)

Dear Editor,

Please find attached the revised version of our manuscript. Following your suggestion, we have made use of professional language editing services. The quality of the English should no longer be a concern.

Regarding your other suggestion "Please keep focus in the manuscript", we thought the best way to take it into consideration would be to rewrite parts of the abstract and, more importantly, the intro. We also provide a methodological flowchart as our first figure in order to improve the flow of the manuscript. We believe the discourse is much clearer now.

Please do not hesitate to let us know if you think there are other specific aspects that could do with improvement.

All the best,

The authors.